# Fast KV Compaction via Attention Matching

**Adam Zweiger** [1]  **Xinghong Fu** [1]  **Han Guo** [1]  **Yoon Kim** [1]

https://github.com/adamzweiger/compaction

## Abstract

Scaling language models to long contexts is often bottlenecked by the size of the key–value (KV) cache. In deployed settings, long contexts are typically managed through *compaction* in token space via summarization. However, summarization can be highly lossy, substantially harming downstream performance. Recent work on Cartridges (Eyuboglu et al., 2025) has shown that it is possible to *train* highly compact KV caches in latent space that closely match full-context performance, but at the cost of slow and expensive end-to-end optimization. This work describes an approach for *fast* context compaction in latent space through **Attention Matching**, which constructs compact keys and values to reproduce attention outputs and preserve attention mass at a per-KV-head level. We show that this formulation naturally decomposes into simple subproblems, some of which admit efficient closed-form solutions. Within this framework, we develop a family of methods that significantly push the Pareto frontier of compaction time versus quality, achieving up to $50\times$ compaction in seconds on some datasets with little quality loss.

## 1. Introduction

Memory has emerged as a critical bottleneck in modern language models (LMs). As such systems are deployed in increasingly long-horizon settings—reasoning, multi-session dialogue, long-running agentic coding—the question of how to efficiently manage, compact, and retrieve contextual information has become paramount. The field is converging on a consensus: models that can remember more, and remember better, will unlock capabilities that remain out of reach for systems constrained by fixed context windows.

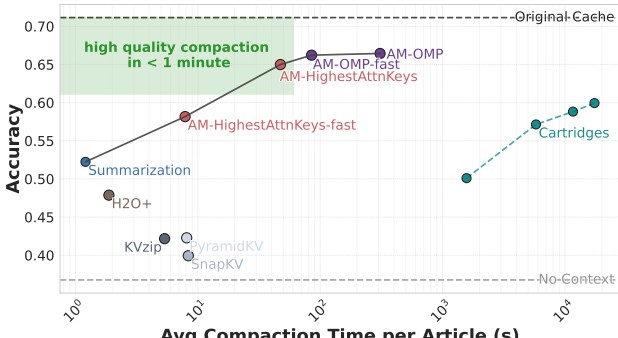

*Figure 1.* **Accuracy vs. Compaction Time Trade-off (Qwen3-4B; QuALITY).** We compare downstream QA accuracy ($n = 894$) after compaction, plotted against the average wall-clock time required to compact a context (seconds, log-scale) using a single H100 GPU at a fixed $50\times$ compaction ratio. Our attention-matching (AM) methods trace a speed–quality tradeoff and form the Pareto frontier, outperforming prior token-selection baselines and exceeding the performance of Cartridges (Eyuboglu et al., 2025) while being 2 orders of magnitude faster; additional Cartridges training may further improve its results.

For autoregressive LMs based on the Transformer architecture, the memory bottleneck is specific: the key-value (KV) cache. Models must retain keys and values from all previous tokens, and in long-context settings, the KV cache can reach many gigabytes per request. Existing approaches to KV cache reduction, such as token eviction (Zhang et al., 2023; Li et al., 2024; Kim et al., 2025b), token merging (Wang et al., 2024; Zhang et al., 2024), and head sparsification (Xiao et al., 2024; 2025), degrade rapidly at high reduction ratios. As a result, real-world systems still rely heavily on summarizing or simply dropping older context (Anthropic, 2025; OpenAI, 2025). Effective *context compaction*[1]—reducing KV cache size in a single pass while preserving downstream model behavior—remains an important open problem.

---
[1]Massachusetts Institute of Technology. Correspondence to: Adam Zweiger <adamz@mit.edu>.

*Proceedings of the 43rd International Conference on Machine Learning*, Seoul, South Korea. PMLR 306, 2026. Copyright 2026 by the author(s).

---
[1]We use "compaction" (instead of "compression") by analogy to log or memory compaction in systems, where many records are consolidated into a smaller representation that preserves the information needed for future access. The term "compaction" is similarly used by agentic coding tools (e.g., Claude Code, OpenAI Codex) to describe this operation, historically implemented purely via summarization.

Recently, Eyuboglu et al. (2025) propose Cartridges, an approach that can be interpreted as performing context compaction in *latent space*. Cartridges uses prefix-tuning (Li & Liang, 2021) on synthetic "self-study" data to *train* a compact KV cache for a given context. This enables compaction ratios as high as $50\times$ on long contexts with minimal performance loss. However, its end-to-end gradient-based optimization strategy can be prohibitively expensive, often requiring several GPU-hours to train a single Cartridge for a given context.

This paper describes a family of methods for fast KV compaction that can achieve Cartridges-level compaction ratios and quality while being orders of magnitude faster. Our approach is based on an **Attention Matching** objective: rather than training a compact KV cache end-to-end on output likelihoods, we directly optimize for compacted keys and values to reproduce the *attention outputs* and *attention mass* for every KV-head in every layer, matching these on a set of reference queries. We show that this Attention Matching objective decomposes into simple subroutines that admit closed-form solutions that are efficient to compute in practice, allowing us to avoid gradient descent entirely at compaction time. Different design choices within our framework yield a family of methods along the speed–performance frontier. Attention Matching enables compaction that is orders of magnitude faster than gradient-based optimization (e.g., minutes rather than hours), with little performance degradation at ratios up to $50\times$ (see Figure 1).

## 2. KV Compaction via Attention Matching

Consider the problem of compacting $T$ tokens from a context. Let $\boldsymbol{K}, \boldsymbol{V} \in \mathbb{R}^{T \times d}$ denote the corresponding keys and values for a single KV-head, where $d$ is the head dimension. Rotary embeddings are assumed to have already been applied to cached keys.

KV compaction aims to replace $(\boldsymbol{K}, \boldsymbol{V})$ with a shorter cache $(\boldsymbol{C}_k, \boldsymbol{C}_v) \in \mathbb{R}^{t \times d}$ with $t < T$ such that conditioning on $(\boldsymbol{C}_k, \boldsymbol{C}_v)$ behaves similarly to conditioning on $(\boldsymbol{K}, \boldsymbol{V})$ for any query $\boldsymbol{q} \in \mathbb{R}^{1 \times d}$.

A key requirement is compatibility with ordinary KV caching: compaction should remain valid even when the compacted prefix is concatenated with arbitrary uncompacted tokens (e.g., the most recent turn) or with future tokens appended after compaction (e.g., user queries or model continuations). Concretely, for any additional key–value blocks $(\boldsymbol{K}_{\text{fixed}}, \boldsymbol{V}_{\text{fixed}})$, we would ideally like the attention output over the compacted prefix to match that over the original prefix (Figure 2). Writing $S = |\boldsymbol{K}_{\text{fixed}}|$ for the number of appended tokens, the denominators below sum over the full concatenated sequence ($T{+}S$ and $t{+}S$ keys, respectively). Written explicitly (omitting the $\sqrt{d}$ scaling

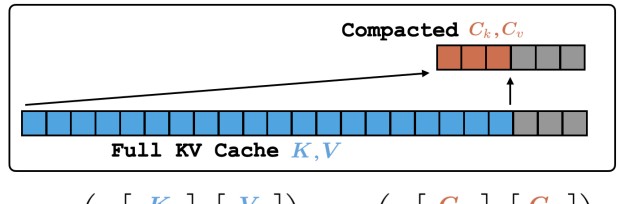

$$\text{Attn}\left(\boldsymbol{q}; \begin{bmatrix} \boldsymbol{K} \\ \boldsymbol{K}_{\text{fixed}} \end{bmatrix}, \begin{bmatrix} \boldsymbol{V} \\ \boldsymbol{V}_{\text{fixed}} \end{bmatrix}\right) \approx \text{Attn}\left(\boldsymbol{q}; \begin{bmatrix} \boldsymbol{C}_k \\ \boldsymbol{K}_{\text{fixed}} \end{bmatrix}, \begin{bmatrix} \boldsymbol{C}_v \\ \boldsymbol{V}_{\text{fixed}} \end{bmatrix}\right)$$

*Figure 2.* **KV compaction via Attention Matching.** We replace the full KV cache $(\boldsymbol{K}, \boldsymbol{V})$ with a smaller cache $(\boldsymbol{C}_k, \boldsymbol{C}_v)$ chosen so that attention outputs are approximately preserved when the compacted prefix is concatenated with arbitrary fixed or future tokens $(\boldsymbol{K}_{\text{fixed}}, \boldsymbol{V}_{\text{fixed}})$.

for readability; see Appendix A), we would like

$$\frac{\exp\left(\boldsymbol{q}\begin{bmatrix} \boldsymbol{K} \\ \boldsymbol{K}_{\text{fixed}} \end{bmatrix}^{\top}\right)\begin{bmatrix} \boldsymbol{V} \\ \boldsymbol{V}_{\text{fixed}} \end{bmatrix}}{\sum_{j=1}^{T+S}\exp\left(\boldsymbol{q}\begin{bmatrix} \boldsymbol{K} \\ \boldsymbol{K}_{\text{fixed}} \end{bmatrix}_j^{\top}\right)} \approx \frac{\exp\left(\boldsymbol{q}\begin{bmatrix} \boldsymbol{C}_k \\ \boldsymbol{K}_{\text{fixed}} \end{bmatrix}^{\top}\right)\begin{bmatrix} \boldsymbol{C}_v \\ \boldsymbol{V}_{\text{fixed}} \end{bmatrix}}{\sum_{j=1}^{t+S}\exp\left(\boldsymbol{q}\begin{bmatrix} \boldsymbol{C}_k \\ \boldsymbol{K}_{\text{fixed}} \end{bmatrix}_j^{\top}\right)}.$$

Directly enforcing this for all possible future $(\boldsymbol{K}_{\text{fixed}}, \boldsymbol{V}_{\text{fixed}})$ is challenging. Instead, we use the fact that attention over concatenated blocks decomposes into a mixture of each block's *locally normalized* attention output, weighted by that block's *attention mass*.[2] For a key block $\boldsymbol{K}$, define

$$\text{Mass}(\boldsymbol{q}; \boldsymbol{K}) = \sum_j \exp\left(\boldsymbol{q}\boldsymbol{K}_j^{\top}\right).$$

This suggests a sufficient compaction strategy that does not depend on unknown future tokens: over a set of reference (or "training") queries, match (i) the compacted block's local attention output and (ii) its attention mass.

One subtlety is that with $t < T$, exact mass matching using only $\boldsymbol{C}_k$ is impossible. For example, for $\boldsymbol{q} = \boldsymbol{0}$ we have $\text{Mass}(\boldsymbol{0}; \boldsymbol{K}) = T$, whereas $\text{Mass}(\boldsymbol{0}; \boldsymbol{C}_k) = t$ for any $\boldsymbol{C}_k \in \mathbb{R}^{t \times d}$. We therefore introduce a per-token scalar bias $\boldsymbol{\beta} \in \mathbb{R}^t$, which multiplicatively reweights the contribution of each retained key to the mass. We later show that these scalar biases arise naturally in Attention Matching.

Specifically, we optimize $(\boldsymbol{C}_k, \boldsymbol{\beta}, \boldsymbol{C}_v)$, where $\boldsymbol{C}_k, \boldsymbol{C}_v \in \mathbb{R}^{t \times d}$ and $\boldsymbol{\beta} \in \mathbb{R}^t$ so that for all queries $\boldsymbol{q}$ of interest,

$$\frac{\exp(\boldsymbol{q}\boldsymbol{K}^{\top})\,\boldsymbol{V}}{\sum_{j=1}^{T}\exp(\boldsymbol{q}\boldsymbol{K}_j^{\top})} \approx \frac{\exp(\boldsymbol{q}\boldsymbol{C}_k^{\top} + \boldsymbol{\beta})\,\boldsymbol{C}_v}{\sum_{j=1}^{t}\exp(\boldsymbol{q}(\boldsymbol{C}_k)_j^{\top} + \boldsymbol{\beta}_j)}, \quad (1)$$

$$\sum_{j=1}^{T}\exp(\boldsymbol{q}\boldsymbol{K}_j^{\top}) \approx \sum_{j=1}^{t}\exp(\boldsymbol{q}(\boldsymbol{C}_k)_j^{\top} + \boldsymbol{\beta}_j). \quad (2)$$

---

[2]Efficient attention implementations such as FlashAttention (Dao et al., 2022) and Cascade Inference (Ye et al., 2024; Juravsky et al., 2024) exploit the same decomposition.

Equation (1) matches the compacted block's local attention output, while Eq. (2) matches its attention mass; together, these preserve the block's contribution under concatenation with arbitrary fixed or future tokens (Appendix A.2). In contrast, methods that only drop or merge tokens without biases (e.g., evicting $T - t$ tokens) systematically underestimate the compacted block's contribution during future decoding.

The scalar biases add negligible memory overhead (an additional factor of $\frac{2d+1}{2d}$) and negligible-to-zero change in attention runtime. They are supported in common attention implementations such as PyTorch SDPA and FlexAttention (Dong et al., 2025).

Finally, although the compacted cache stores only $t$ KV entries, it retains a *logical length* $T$ so that newly appended tokens receive the same position IDs (and thus the same RoPE phases) as they would under the uncompacted prefix. We view this as disentangling the cumulative length a cache has *seen* from its physical size.

## 3. Methods

We now present a family of methods for constructing $(C_k, \boldsymbol{\beta}, C_v)$. Our approach is defined with respect to a set of *reference queries* $\boldsymbol{Q}_{\text{ref}} = [\boldsymbol{q}_1; \ldots; \boldsymbol{q}_n] \in \mathbb{R}^{n \times d}$, which serve as a proxy for the queries the model is likely to produce when attending to the context.

Jointly optimizing $(C_k, \boldsymbol{\beta}, C_v)$ is computationally difficult and typically requires gradient-based optimization, which we aim to avoid at compaction time. Instead, we first construct the compacted keys $C_k$, then compute the bias terms $\boldsymbol{\beta}$, and finally obtain $C_v$, all in closed form.

### 3.1. Sampling Reference Queries $Q_{\text{ref}}$

We consider two main approaches for constructing $\boldsymbol{Q}_{\text{ref}}$.

**Repeat-prefill.** Following KVzip (Kim et al., 2025b), we construct a sequence

"{$\mathcal{C}$} Repeat the previous context. {$\mathcal{C}$}"

(with the model's chat template applied appropriately), run a prefill pass on this sequence, and extract the query vectors used while the model reconstructs $\mathcal{C}$ (i.e., query activations starting from the instruction through the second $\mathcal{C}$). We found that the simpler variant of running a prefill on $\mathcal{C}$ alone, which we call context-prefill (and used by H2O (Zhang et al., 2023)), was cheaper (see Table 3) but performed slightly worse than repeat-prefill.

**Self-study.** Self-study (Eyuboglu et al., 2025) generates synthetic interactions conditioned on a fixed context $\mathcal{C}$. We use it as a lightweight way to broaden the query distribution: we prompt the model with $\mathcal{C}$ and a small set of fixed prompts

(e.g., "Aggregate all key facts mentioned in the context") or model-generated conversation-starters. We then sample responses and extract the resulting query vectors. In practice, we run only four prompts (Appendix C.4), with the majority of reference queries coming from repeat-prefill.

These approaches concentrate reference queries on representations the model naturally produces when processing and reasoning about $\mathcal{C}$. Empirically, in Appendix D, we observe that self-study yields the best downstream performance; context-prefill and repeat-prefill are nearly as good while being faster. We also find that randomly sampling $\boldsymbol{q}_i \sim \mathcal{N}(0, \boldsymbol{I}_d)$ (random-vectors) works, though it lags the other approaches.

**"On-policy" queries.** Per-layer compaction can induce distribution shift in queries: compacting early layers changes the residual stream seen by later layers, so the queries they produce may differ from those extracted from the unmodified model. To reduce this mismatch, we compact layers sequentially and, for each layer $\ell$, extract $\boldsymbol{Q}_{\text{ref}}^\ell$ by running the model with layers $< \ell$ already compacted, then optimize compaction at layer $\ell$ using these on-policy queries. This yields slight but consistent improvements.

### 3.2. Constructing $\beta$ and $C_v$

Before describing how we construct compact keys $C_k$, we first explain how we learn $\boldsymbol{\beta}$ and $C_v$ to satisfy the attention-matching objectives (Eqs. 1–2) given reference queries $\boldsymbol{Q}_{\text{ref}} = [\boldsymbol{q}_1; \ldots; \boldsymbol{q}_n] \in \mathbb{R}^{n \times d}$ and compacted $C_k$. (In one of the variants we consider, we alternate between fitting subsets of $C_k$ and $\boldsymbol{\beta}, C_v$.)

**Fitting $\beta$.** Let $\mathbf{m} = [m_1, \ldots, m_n]^\top$ be the vector of original attention mass, i.e.,

$$m_i = \sum_{k=1}^T \exp(\boldsymbol{q}_i \boldsymbol{K}_k^\top).$$

Parameterizing $w_j = \exp(\boldsymbol{\beta}_j) > 0$, we would then like

$$m_i \approx \sum_{j=1}^t \exp\big(\boldsymbol{q}_i (\boldsymbol{C}_k)_j^\top + \boldsymbol{\beta}_j\big) = \sum_{j=1}^t w_j \exp\big(\boldsymbol{q}_i (\boldsymbol{C}_k)_j^\top\big).$$

This results in the following optimization problem

$$\min_{\mathbf{w}:\, w_j \geq 0} \|\boldsymbol{A}\mathbf{w} - \mathbf{m}\|_2^2$$

where $\boldsymbol{A}_{ij} = \exp\big(\boldsymbol{q}_i (\boldsymbol{C}_k)_j^\top\big)$ and $\mathbf{w} = [w_1, \ldots, w_t]^\top$. We solve for this via nonnegative least squares (NNLS; see Appendix C.2 for implementation details), and then set $\boldsymbol{\beta}_j = \log(w_j)$, clamping $w_j$ to a small positive value if needed. Intuitively, $w_j$ represents how many original keys' worth of attention mass the compact key $(\boldsymbol{C}_k)_j$ accounts for.

**Fitting $C_v$.** Recall that the attention-output matching condition for a query $q$ is given by

$$\frac{\exp(qK^\top)V}{\sum_{j=1}^{T}\exp(qK_j^\top)} \approx \left(\frac{\exp(qC_k^\top + \beta)}{\sum_{j=1}^{t}\exp(q(C_k)_j^\top + \beta_j)}\right)C_v$$

With $C_k$ and $\beta$ fixed, we can solve for $C_v$ with ordinary least squares. For each reference query $q_i$ we compute

$$y_i = \frac{\exp(q_iK^\top)V}{\sum_{j=1}^{T}\exp(q_iK_j^\top)} \in \mathbb{R}^{1\times d},$$

$$x_i = \frac{\exp(q_iC_k^\top + \beta)}{\sum_{j=1}^{t}\exp(q_i(C_k)_j^\top + \beta_j)} \in \mathbb{R}^{1\times t}.$$

Stacking into $Y = [y_1; \ldots; y_n] \in \mathbb{R}^{n\times d}$ and $X = [x_1; \ldots; x_n] \in \mathbb{R}^{n\times t}$, we solve for $C_v$ with

$$C_v^\star = \arg\min_{C_v}\|XC_v - Y\|_F^2, \tag{3}$$

$$= (X^\top X)^{-1}X^\top Y. \tag{4}$$

In summary, given any set of compacted keys $C_k$, simple linear-algebra routines—least squares and nonnegative least squares—allow us to learn $(\beta, C_v)$ that minimize $\ell_2$ error in the attention mass and attention outputs over the reference queries. Now the main challenge becomes selecting $C_k$.

### 3.3. Selecting $C_k$

We do not have a closed-form solution for constructing $C_k$ in general. However we found it empirically effective (and efficient) to restrict $C_k$ to be a subset of the original keys, i.e., $C_k = K_{S,:}$ for some index set $S \subset \{1, \ldots, T\}$ with $|S| = t$. This allows us to avoid iterative gradient-based optimization. We consider two methods for choosing this subset, which have different efficiency-performance tradeoffs.

**Highest attention keys.** A simple approach in prior work on KV cache eviction/pruning (Zhang et al., 2023; Li et al., 2024; Kim et al., 2025b) is to retain keys that receive the most attention. In our setting, we measure this under the reference queries. For each reference query $q_i$, we compute attention weights over the original keys:

$$a_i = \text{softmax}(q_iK^\top) \in \mathbb{R}^{1\times T}.$$

We then aggregate these weights across queries via root mean square over $(a_{1,j}, \ldots, a_{n,j})$ to obtain a per-key importance score $s_j$. We found RMS to be more robust than mean or max aggregation (Appendix F.1).

**Orthogonal matching pursuit (OMP) keys.** The method above is a fast and effective heuristic in practice. As a more direct alternative, we can explicitly match the attention mass

---

**Algorithm 1** OMP Key Selection

**Require:** Original keys $K \in \mathbb{R}^{T\times d}$, queries $Q \in \mathbb{R}^{n\times d}$, budget $t$
**Ensure:** Indices $S$, weights $\mathbf{w}$ (with $\beta = \log \mathbf{w}$)
1: $\Phi_{ij} \leftarrow \exp(q_iK_j^\top/\sqrt{d})$ {Mass feature matrix}
2: $m_i \leftarrow \sum_{j=1}^{T}\Phi_{ij}$ {Target mass vector}
3: $\mathbf{r} \leftarrow \mathbf{m}, \quad S \leftarrow \emptyset$
4: **for** $k = 1$ **to** $t$ **do**
5: $\quad j^\star \leftarrow \arg\max_{j\notin S}(\mathbf{r}^\top\Phi_{:,j})$
6: $\quad S \leftarrow S \cup \{j^\star\}$
7: $\quad \mathbf{w} \leftarrow \arg\min_{\mathbf{w}\geq 0}\|\Phi_{:,S}\mathbf{w} - \mathbf{m}\|_2^2$ {NNLS}
8: $\quad \mathbf{r} \leftarrow \mathbf{m} - \Phi_{:,S}\mathbf{w}$
9: **end for**
10: **return** $S, \mathbf{w}$

---

using orthogonal matching pursuit (OMP; Tropp & Gilbert, 2007), which greedily builds $C_k$ and $\beta$ to best satisfy Eq. 2.

Define the mass feature matrix $\Phi \in \mathbb{R}^{n\times T}$ with $\Phi_{ij} = \exp(q_iK_j^\top)$ and target vector $\mathbf{m}$ with $m_i = \sum_j \Phi_{ij}$. We seek a sparse subset $S$ and weights $\mathbf{w} \geq 0$ minimizing $\|\Phi_{:,S}\mathbf{w} - \mathbf{m}\|_2^2$. OMP selects $S$ greedily: at each step, it adds the key whose column maximally reduces the residual, then refits $\mathbf{w}$ via NNLS (Algorithm 1). This algorithm directly yields $\beta = \log \mathbf{w}$; we then fit $C_v$ via least squares as above.

While OMP performs best empirically, it is slower than the other methods: greedy selection with NNLS refitting scales at least linearly in $t$. In practice, selecting multiple keys per step and refitting at intervals reduces compaction time by $4$–$8\times$ with little degradation (Appendix C.1).

### 3.4. Nonuniform Compaction

Different attention heads can exhibit different attention patterns (Wu et al., 2025; Xiao et al., 2025; Bick et al., 2025), suggesting that a fixed compaction ratio across all heads is suboptimal. We say that *uniform* compaction uses the same ratio for every KV-head at every layer, whereas *nonuniform* compaction assigns a potentially different ratio to each head and layer.[3]

To motivate a reusable nonuniform compaction schedule for each model, we first show that attention-head sensitivity is largely input-invariant—although absolute loss varies by

---

[3] A naïve nonuniform cache implementation would pad all heads to the length of the longest KV-head, increasing the effective context length and partially negating compute savings. However, attention kernels that support variable-length sequences (e.g., FlashAttention (Dao, 2024)) can avoid this overhead by packing per-head KV segments into a flat, variable-length (varlen) representation. With such packing, nonuniform KV caches can achieve the same memory and compute benefits of a uniform cache of the same total size (Feng et al., 2025; Rehg, 2024).

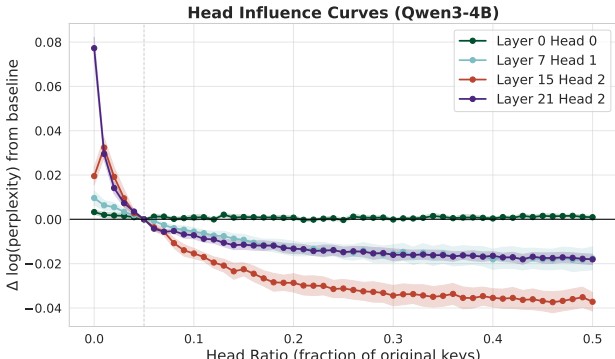

*Figure 3.* **Head sensitivity curves in Qwen3-4B.** We fix all KV heads to a baseline compaction ratio of $0.05\times$ and vary the compaction ratio of a single head. We report the change in loss relative to the baseline (lower is better) as a function of the varied head's compaction ratio. Curves are averaged over 10 QuALITY articles; shaded regions denote $\pm 1$ standard error of the mean across articles. Some heads (e.g., `L0H0`) are largely insensitive to additional capacity, whereas others (e.g., `L15H2`) benefit substantially from storing more KV pairs.

example, the relative ranking of head importance remains stable. Figure 3 shows the resulting sensitivity curves for Qwen3-4B, averaged over contexts. These rankings also transfer across datasets; the same qualitative pattern appears on LongHealth (Appendix E).

**Precomputed head budgets.** This stability enables us to precompute, once per model, a nonuniform compaction schedule—a per-head *share* of the total KV budget—that can be reused across contexts and compaction ratios. Concretely, given the per-head sensitivity curves, we solve a discrete resource allocation problem using a standard greedy exchange algorithm (Algorithm 4). Starting from a uniform allocation across heads, we iteratively swap units of KV budget between heads to minimize loss implied by the sensitivity curves. We repeat this process until no swap yields further improvement. This procedure assumes approximate separability across heads: we predict the effect of reallocating budget using single-head sensitivity curves measured with other heads held fixed.

The resulting schedule is an efficient model-specific nonuniform allocation procedure that prioritizes capacity for heads most sensitive to compaction. We found this allocation to be robust across instances/datasets, and thus this procedure only needs to be performed once for a given model. A visualization of the learned head budgets is provided in Appendix E.

### 3.5. Chunked Compaction

To support long contexts, we apply compaction independently to contiguous chunks of the input. Because Attention Matching can select and compact arbitrary token subsets,

each chunk can be processed separately and the resulting KV caches concatenated to form a single compacted cache.

We consider two implementations. In *KV-based* chunking, we prefill the full context, slice out the KV states corresponding to each chunk, compact them independently, and merge the compacted chunks. In *text-based* chunking, we instead prefill and compact each chunk in isolation, and then align the resulting keys to their original global positions via a RoPE phase shift before merging. Detailed descriptions of both variants are given in Appendix C.3.

The text-based approach is an approximation because chunks are prefilled without cross-chunk interactions. In practice, we find that KV-based chunking more faithfully preserves model behavior, even when chunks are semantically independent, and therefore use it by default in all experiments, including baselines.

## 4. Results

We evaluate Attention Matching across a range of compaction strategies, measuring downstream quality and compaction efficiency on long-context benchmarks.

**Compared Variants.** We evaluate four representative compaction variants that trade off compaction time and downstream quality by varying how reference queries are obtained and how keys are selected. The variants are ordered by performance:

- **AM-OMP:** On-policy $Q_{\text{ref}}$ from self-study + repeat-prefill; fit $(C_k, \beta)$ jointly via OMP; fit $C_v$ by least squares.

- **AM-OMP-fast:** Same as AM-OMP, but with OMP speedups ($k = 4$ keys selected per iteration, $\tau = 2$ iterations between NNLS refits; Appendix C.1).

- **AM-HighestAttnKeys:** On-policy $Q_{\text{ref}}$ from self-study + repeat-prefill; select $C_k$ by highest attention; fit $\beta$ via NNLS; fit $C_v$ by least squares.

- **AM-HighestAttnKeys-fast:** Same as AM-HighestAttnKeys, but with $Q_{\text{ref}}$ coming only from repeat-prefill.

With self-study + repeat-prefill, we use at most 50,000 reference queries per KV-head for each context/chunk. Importantly, this does not require a large number of generated tokens: under GQA, each token yields multiple query vectors per KV-head, and the majority of reference queries are obtained via repeat-prefill rather than self-study. On QuALITY, we generate on average approximately 5k self-study tokens and 7k repeat-prefill tokens per context.

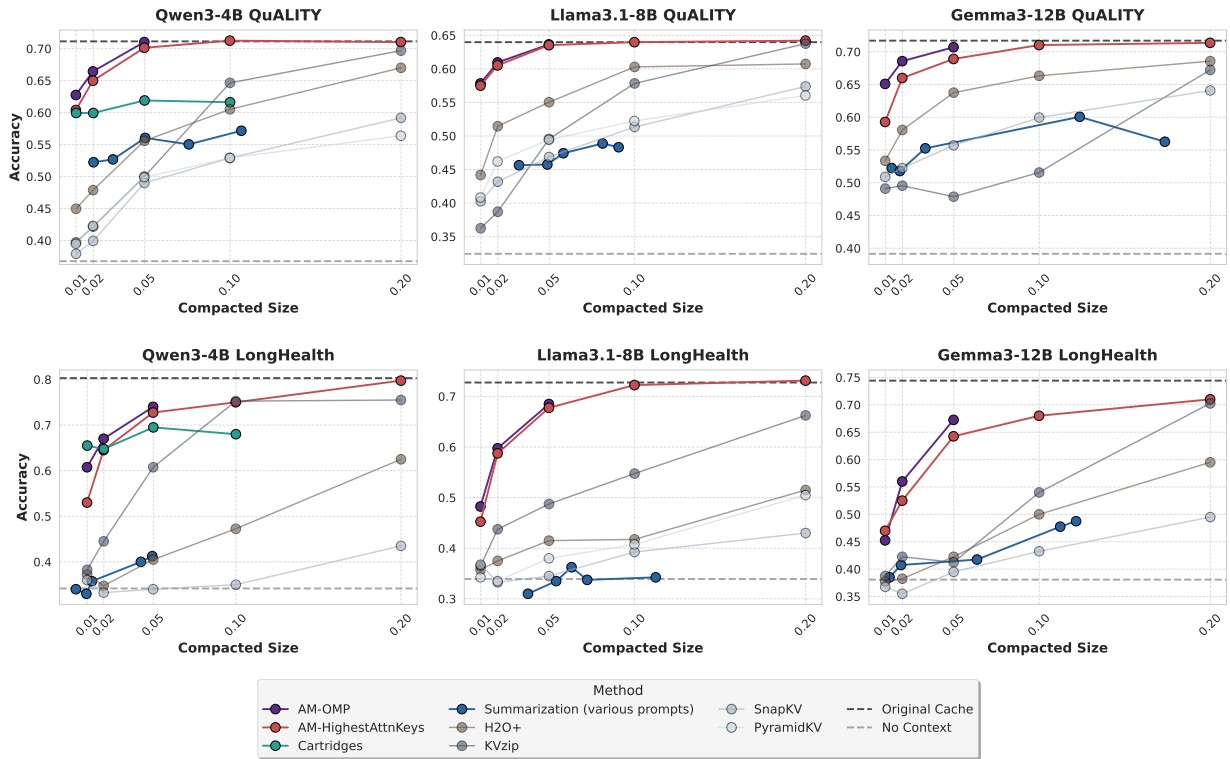

*Figure 4.* **Accuracy vs. compaction ratio across methods.** We compare AM-OMP and AM-HighestAttnKeys against Cartridges, summarization, and four prior methods. Evaluations are conducted on QuALITY and LongHealth using Qwen3-4B, Llama3.1-8B, and Gemma3-12B. Attention Matching (AM) consistently outperforms other approaches across compaction ratios, while matching Cartridges' performance at ultra-high compaction.

**Baselines.** We compare against (i) **Cartridges** (Eyuboglu et al., 2025), which performs end-to-end gradient-based optimization of a compact latent KV cache for each context; (ii) baselines based on token pruning using attention scores (H2O (Zhang et al., 2023), SnapKV (Li et al., 2024), PyramidKV (Cai et al., 2025), KVzip (Kim et al., 2025b)) and a token-merging method KVMerger (Wang et al., 2024), instantiated in our one-shot, question-agnostic setting as described in Appendix B.3; and (iii) **summarization**, which replaces the original context with a shorter textual summary, either produced for the full document or by summarizing chunks and concatenating the summaries (Table 7).

**Evaluation protocol.** We evaluate on QuALITY (Pang et al., 2022), a long-document comprehension benchmark (5–8k tokens; 15–20 questions per context), and LongHealth (Adams et al., 2024), a highly information-dense patient-records QA task (60k tokens per context; 100 questions per context, where each context aggregates 5 patients; 4 contexts total). See Appendix B.2 for a sample context and question.

For each context, we compact once and then answer all associated multiple-choice questions by decoding from the compacted cache. For all methods, we keep the chat tem-

plate tokens fixed during compaction (e.g., BOS / system-prefix tokens, which can act as attention sinks (Xiao et al., 2024)) and re-attach them after compaction. We report the compacted size $t/T$ with respect to the $T$ article tokens, excluding chat-template tokens. While we found our method is robust to compacting these template tokens, we fix them for a fair comparison with baselines. We apply KV-based chunked compaction on LongHealth with 5 chunks for all methods except for Cartridges.

Our QuALITY evaluation is done over 50 contexts. Since evaluating Cartridges was computationally expensive, we only evaluate it on Qwen, on a fixed 20-context subset selected a priori; scores on this subset showed little deviation from the full set of 50 (Appendix B.2).

### 4.1. Main Results

We first analyze the trade-off between compaction time and downstream accuracy. Figure 1 plots this for Qwen3-4B on QuALITY at a fixed $50\times$ ratio. (See Figure 11 in the appendix for perplexity results.) Our Attention Matching (AM) methods trace the Pareto frontier and bridge the gap between fast heuristic baselines (e.g., summarization, H2O+, KVzip), which degrade significantly at this ratio,

| Method | 0% | 1% | 2% | 5% | 10% | 20% | 100% |
|---|---|---|---|---|---|---|---|
| *Qwen3-4B* | | | | | | | |
| Baseline (no ctx / orig.) | 0.104 | | | | | | **0.444** |
| Summarization | | | | 0.219 | 0.283 | | |
| H2O+ | | 0.188 | 0.243 | 0.310 | 0.350 | 0.390 | |
| KVMerger | | 0.171 | 0.215 | 0.260 | 0.303 | 0.356 | |
| KVzip | | 0.127 | 0.124 | 0.220 | 0.411 | **0.438** | |
| AM-HighestAttnKeys | | **0.338** | **0.373** | **0.416** | **0.428** | 0.435 | |
| *Llama-3.1-8B-Instruct* | | | | | | | |
| Baseline (no ctx / orig.) | 0.105 | | | | | | **0.456** |
| Summarization | | | | 0.253 | 0.280 | | |
| H2O+ | | 0.199 | 0.243 | 0.301 | 0.348 | 0.372 | |
| KVzip | | 0.102 | 0.104 | 0.168 | 0.347 | 0.422 | |
| AM-HighestAttnKeys | | **0.317** | **0.387** | **0.436** | **0.452** | **0.450** | |

*Table 1*. **QASPER, average token-level F1** (562 contexts, 2,010 questions).

and optimization-heavy methods like Cartridges.

In Figure 4, we sweep the compaction ratio across 3 models on QuALITY and LongHealth. We observe that Attention Matching methods consistently outperform token-eviction baselines and summarization, particularly in the high-compaction regime ($20\times$–$100\times$). Performance degrades faster on LongHealth, which is highly information-dense (Appendix B.2). Summarization does particularly poorly on this dataset, matching the no-context baseline. KVzip sometimes matches Attention Matching performance, which we attribute to its non-uniform budget outperforming our precomputed budget at certain compaction ratios.

### 4.2. Generalization to Other Benchmarks

To test whether Attention Matching generalizes beyond multiple-choice comprehension, we evaluate on QASPER (Dasigi et al., 2021), a free-form generative QA task over NLP papers (token-level F1 over 562 contexts and 2,010 questions), and LongBench v2 (Bai et al., 2025), which spans code repositories, tables, multilingual documents, and long structured contexts. On these benchmarks we additionally compare against KVMerger (Wang et al., 2024), a token-*merging* baseline that collapses clusters of similar adjacent keys into a single averaged KV pair rather than evicting tokens; we run it in the same one-shot, question-agnostic protocol as our other baselines (Appendix B.3). For LongBench v2, absolute accuracy is low for these base models, so we report accuracy on the 47-context subset the model answers correctly with the full context but incorrectly without it, re-evaluating every method so all numbers are comparable (Appendix B.2). Because the subset is selected on a separate run and we decode with temperature sampling, the 0% (no-context) and 100% (full-context) accuracies on this subset are not exactly 0% and 100%. Tables 1 and 2 show that Attention Matching again achieves the strongest performance, retaining most of the full-context score even at 1% cache ($100\times$ compaction), where the baselines collapse toward the no-context score. Appendix F.4 reports further results, including the retrieval-

| Method | 0% | 1% | 2% | 5% | 10% | 20% | 100% |
|---|---|---|---|---|---|---|---|
| Baseline (no ctx / orig.) | 17.0 | | | | | | 72.3 |
| Summarization | | | 23.4 | 29.8 | 44.7 | | |
| H2O+ | | 10.6 | 21.3 | 36.2 | 44.7 | 61.7 | |
| KVMerger | | 19.2 | 21.3 | 34.0 | 44.7 | 48.9 | |
| KVzip | | 17.0 | 21.3 | 36.2 | 48.9 | 66.0 | |
| AM-HighestAttnKeys | | **61.7** | **59.6** | **66.0** | **70.2** | **68.1** | |

*Table 2*. **LongBench v2 accuracy (%)**, Qwen3-4B-Instruct-2507, 47-context subset (correct with full context, incorrect without; max 97k, avg 42k tokens).

intensive RULER benchmark (Hsieh et al., 2024), where even our fastest variant surpasses KVzip and six methods from the public `kvpress` leaderboard.

### 4.3. Sliding Window Attention

The effectiveness of nonuniform compaction schedules gives some insight into why hybrid architectures with *sliding-window attention* (Child et al., 2019; Beltagy et al., 2020) and only a few global layers can retain much of the performance of full-attention models: even when trained with full attention, many KV-heads naturally specialize to modeling local context, while only a small number of globally attending heads capture the long-range dependencies.

Does this imply that compaction is less effective for these hybrid models? To verify that our improvements still hold on sliding-window models, we evaluate Gemma-3-12B, which has an aggressive 5:1 ratio of sliding-window layers to global layers. We compact only the global-attention layers, leaving the sliding-window layers unchanged. Accordingly, the reported compaction ratios are computed over only the global-attention portion of the KV cache. As shown in Figure 4, we require only slightly more conservative compaction ratios compared to the two full-attention models.

### 4.4. Compaction Efficiency

We profile the wall-clock compute cost of each component of our method on a 60k-token LONGHEALTH context using Gemma-3-12B (64 full-attention KV heads; head dimension $d_{\text{head}}=256$); see Table 3. All timings are measured on a single H200 GPU. We compute compacted keys and values in FP32 and then cast to BF16 for storage; we do not experiment with quantization in this work.

Query generation dominates runtime in this long-context setting and can likely be substantially optimized. All results here use a single GPU and self-study/repeat-prefill with chunked prefill with chunk size 4096.

Although we only evaluate models up to 12B parameters, we note that KV cache size varies quite little across different size variants of open-source models. For example, for the same sequence length, Qwen3-235B-A22B yields a KV cache only $\sim$30% larger than Qwen3-4B (96,256 vs. 73,728 elements per token). We therefore expect our compaction efficiency results to scale without much additional cost to

| Stage | Method | Time (s) |
|---|---|---|
| | context-prefill | 7 |
| Query generation | repeat-prefill | 8 |
| | self-study | 139 |
| | Highest attention | 3 |
| Key selection | OMP | 565 |
| | OMP-fast ($k=4$, refit interval=2) | 104 |
| $\beta$ fitting | NNLS | 2.2 |
| Value fitting | Least squares | 1.8 |

*Table 3.* Wall-clock compaction time breakdown on a 60k-token LONGHEALTH context with Gemma-3-12B on a *single H200 GPU*.

| Method | Cache Size (%) | Accuracy (%) |
|---|---|---|
| Full Context | 100.00 | 71.1 |
| Summarize | 4.80 | 55.2 |
| Summarize × (0.2× AM-OMP) | 0.92 | 55.7 |
| Summarize × (0.1× AM-OMP) | 0.46 | 55.0 |
| Summarize × (0.05× AM-OMP) | 0.21 | 49.2 |

*Table 4.* Summarization plus Attention Matching. Applying AM-OMP on top of a summary enables up to ~200× total compaction with accuracy comparable to summarization alone.

much larger mixture-of-experts models commonly used in deployment.

### 4.5. Extension: Summarization plus Attention Matching

While summarization rapidly degrades performance on tasks that require extracting knowledge and reasoning across long contexts (Section 4.1), it may be acceptable in cases where not all prior information needs to be retained (for example, when there exist retrieval or search mechanisms), or for filtering out noise or hallucinations.

In Table 4, we demonstrate that Attention Matching works on top of summarization: applying AM-OMP to summarized text achieves 200× compaction (6340→31 effective tokens on average) with performance comparable to summarization alone, which in our case provides only ~ 20× compaction.

### 4.6. Ablations

We've introduced several concepts: post-compaction attention biases, fitted values, nonuniform head budgets, and query-sampling methods. In Figure 5, we leave out components of our main AM-OMP method and measure loss on reference generations. We see that every component is necessary to achieve the best results, with the least important being the use of self-study and the use of on-policy queries, and the most important being nonuniform head budgets.

We note that omitting attention biases and retaining the original values for selected keys still yields a reasonable approximation. However, refitting $\beta$ and $C_v$ via (nonnegative)

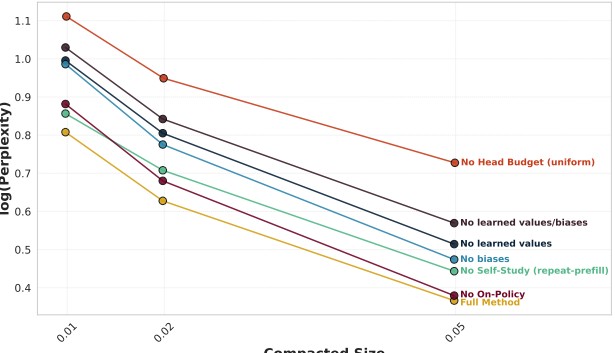

*Figure 5.* **Leave-one-out experiments.** We ablate our main AM-OMP method and measure average log-perplexity of generations decoded from the original cache, since this is lower in variance and well-correlated with downstream performance (Appendix F.2). In "no biases," we first compute OMP as usual and then zero out $\beta$, keeping the keys selected by OMP.

least squares better aligns with our objective of approximating the original cache, adds little overhead, and consistently improves performance.

## 5. Related Work

While some prior works approach KV cache reduction as a per-token online problem—continuously evicting or merging tokens at each decoding step to maintain a fixed state size (e.g., Zhang et al., 2023; Oren et al., 2024)—we study compaction as a one-shot operation applied at the moment a context becomes too large. Methods in this setting are able to make globally informed decisions over the full prefix. Moreover, one-shot compaction can be applied repeatedly to maintain a fixed maximum state size, as in Appendix F.3.

Token-space approaches to handling long context generally involve combinations of retrieval-augmented generation (Lewis et al., 2020), context compaction via summarization or token dropping (Jiang et al., 2023; Li et al., 2023; Anthropic, 2025), and agentic memory management (Packer et al., 2024; Zhang et al., 2025). These approaches are largely orthogonal to latent-space compaction. While summarization rapidly degrades performance on tasks that require extracting knowledge and reasoning across long contexts (Section 4), it may be acceptable in cases where not all prior information needs to be retained (for instance, when there exist retrieval or search mechanisms). The two approaches can be combined with strong performance (Table 4); the most effective memory system may have a mix of retrieval and compaction in both token-space and latent-space.

In addition to the baselines that we evaluate (Zhang et al., 2023; Li et al., 2024; Cai et al., 2025; Kim et al., 2025b), a large body of other work explores token-merging (Wang et al., 2024; Zhang et al., 2024; Liu et al., 2025; Wan et al.,

2025) and token-eviction (Oren et al., 2024; Ge et al., 2024; Tang et al., 2024; Chari & Durme, 2025; Łańcucki et al., 2025). The KVPress (Devoto et al., 2025) repository benchmarks over 20 such methods; we include the top-performing approach, KVzip (Kim et al., 2025b), as well as the token-merging method KVMerger (Wang et al., 2024), among our baselines (Appendix B.3).

One line of prior work explores having models *generate* a set of soft tokens or "gist" tokens (Bulatov et al., 2022; Chevalier et al., 2023; Mu et al., 2023) that can be used to replace the original cache. These methods require training or finetuning the model to produce such representations, whereas our approach operates as a post-hoc procedure on any pretrained model without modification.

Our work is also related to Lexico (Kim et al., 2025a), which compresses keys and values by learning a *universal* dictionary per layer and uses orthogonal matching pursuit (OMP) at inference time to store each per-token key/value vector as a sparse code over the learned atoms. Lexico's objective is $\ell_2$ reconstruction of KV vectors whereas we optimize compacted KV vectors for matching attention behavior.

DuoAttention (Xiao et al., 2025) proposes that attention heads be split into a small set of *retrieval* heads that use a full-length KV cache, and *streaming* heads that can run with a constant-length cache containing only attention sinks and recent tokens; they learn this partition via gradient optimization over per-head *gates* that blend full vs. streaming attention on synthetic data, then binarize the gates for deployment. Our nonuniform compaction shares the insight that heads vary in their sensitivity to KV capacity, but we learn continuous per-head budgets via *sensitivity curves* and compact all heads to varying degrees.

The notion of adding a per-token scalar attention bias was suggested in T5 (Raffel et al., 2020) and ALiBi (Press et al., 2022) as a form of positional encoding. Our method also employs attention biases, but for a different purpose: to correct for the attention mass received by the compacted cache.

## 6. Discussion

Disentangling logical KV length from physical cache size yields a practical primitive operation: KV compaction, which can be invoked on arbitrary portions of context whenever a sequence grows too large. Attention Matching is a natural paradigm for KV compaction because it approximately preserves the two ingredients that determine a block's contribution under concatenation—attention outputs and attention mass. Moreover, Attention Matching enables efficient algorithms for finding compact keys and values.

Our work still has several limitations. For one, while our approach is orders of magnitude faster than end-to-end gradient-based methods (i.e., Cartridges), our unoptimized OMP + self-study variants still required several minutes for compaction. And while our approach outperforms Cartridges in terms of accuracy at $50\times$ compaction rates, Cartridges outperforms our approach at more extreme ($100\times$) compaction rates on some benchmarks (LongHealth). We attribute this to the fact that gradient-based optimization can search over a wider space of compact representations—in particular, it is not restricted to optimizing intermediate attention outputs or selecting keys from the original cache—which becomes increasingly important as the compaction budget shrinks.

**Future work.** Beyond speeding up query-generation or OMP-style selection, a promising direction is to move away from subset selection for $C_k$ (e.g., directly optimizing compact keys). Another interesting direction is architectures or training procedures that support compaction as a simple primitive, or explicitly operate over a fixed set of keys and values.

Integrating KV compaction into inference engines (e.g., RadixAttention-style prefix caching, varlen KV packing, and disaggregated compaction; Appendix G.3) and combining token-space retrieval/summarization with latent-space compaction are important systems directions that still require significant effort.

Finally, while our experiments focus on one-shot compaction of a fixed context, the problem of maintaining memory over long-horizon interactions is another very natural setting for KV compaction. A promising application is *online compaction*, where we compact the KV cache mid-trajectory to support arbitrarily long-horizon generation under a fixed physical memory budget. Appendix F.3 provides a preliminary result showing that repeated context compaction via Attention Matching preserves reasoning performance on AIME. This is especially promising for long-horizon agentic settings such as coding agents, where tool call outputs and execution traces can take up a significant amount of context. Online latent compaction could be a principled alternative or complement to turn-dropping and summarization.

## 7. Conclusion

We study Attention Matching as an objective for fast latent-space compaction. Our methods significantly improve the Pareto frontier of compaction cost versus quality.

## Acknowledgements

We would like to thank Ani Nrusimha, Alicia Li, Sabri Eyuboglu, Zifan Wang, Jyo Pari, Oliver Sieberling, and Tarushii Goel for their valuable discussions and feedback.

This work was supported by the National Science Foundation under CAREER Award No. 2441872, MIT-IBM Watson AI Lab, and the Prof. JCR Licklider Endowed UROP Fund for MIT UROP research.

## Impact Statement

This paper presents work whose goal is to advance the field of Machine Learning. There are many potential societal consequences of our work, none which we feel must be specifically highlighted here.

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

_files/paper/2023/file/6ceefa7b15572
587b78ecfcebb2827f8-Paper-Conference.
pdf.

## A. Attention Matching Details

This appendix restates the attention-matching objectives from Section 2 in more detail.

### A.1. Notation

Fix a single layer and KV-head with head dimension $d$. Let the original keys and values be

$$\boldsymbol{K} \in \mathbb{R}^{T \times d}, \qquad \boldsymbol{V} \in \mathbb{R}^{T \times d},$$

and let the compacted keys, values, and per-token bias be

$$\boldsymbol{C}_k \in \mathbb{R}^{t \times d}, \qquad \boldsymbol{C}_v \in \mathbb{R}^{t \times d}, \qquad \boldsymbol{\beta} \in \mathbb{R}^t.$$

Let $\boldsymbol{Q}_{\text{ref}} \in \mathbb{R}^{n \times d}$ denote the reference query matrix whose $i$th row is $\boldsymbol{q}_i$.

**Attention operators.** For any key/value block $(\boldsymbol{K}, \boldsymbol{V})$ and query $\boldsymbol{q}$, define the scaled logits

$$\ell(\boldsymbol{q}; \boldsymbol{K}) = \tfrac{1}{\sqrt{d}} \boldsymbol{q} \boldsymbol{K}^\top$$

and the unnormalized mass

$$\text{Mass}(\boldsymbol{q}; \boldsymbol{K}) = \sum_j \exp(\ell(\boldsymbol{q}; \boldsymbol{K})_j).$$

Define the locally normalized attention output

$$\text{Attn}(\boldsymbol{q}; \boldsymbol{K}, \boldsymbol{V}) = \frac{\exp(\ell(\boldsymbol{q}; \boldsymbol{K})) \, \boldsymbol{V}}{\text{Mass}(\boldsymbol{q}; \boldsymbol{K})}.$$

For compacted parameters $(\boldsymbol{C}_k, \boldsymbol{\beta}, \boldsymbol{C}_v)$ we use

$$\ell(\boldsymbol{q}; \boldsymbol{C}_k, \boldsymbol{\beta}) = \tfrac{1}{\sqrt{d}} \boldsymbol{q} \boldsymbol{C}_k^\top + \boldsymbol{\beta}, \quad \text{Mass}(\boldsymbol{q}; \boldsymbol{C}_k, \boldsymbol{\beta}) = \sum_j \exp(\ell(\boldsymbol{q}; \boldsymbol{C}_k, \boldsymbol{\beta})_j),$$

and

$$\text{Attn}(\boldsymbol{q}; \boldsymbol{C}_k, \boldsymbol{\beta}, \boldsymbol{C}_v) = \frac{\exp(\ell(\boldsymbol{q}; \boldsymbol{C}_k, \boldsymbol{\beta})) \, \boldsymbol{C}_v}{\text{Mass}(\boldsymbol{q}; \boldsymbol{C}_k, \boldsymbol{\beta})}.$$

### A.2. Mass-preserving Attention Matching

We justify replacing attention over $[\boldsymbol{K}; \boldsymbol{K}_{\text{fixed}}]$ with the two objectives (1)–(2). However, we observe that attention over 2 subsets of indices decomposes into a mixture whose weights are determined by *unnormalized attention mass*.

Let $(\boldsymbol{K}, \boldsymbol{V})$ be the block we compact and $(\boldsymbol{K}_{\text{fixed}}, \boldsymbol{V}_{\text{fixed}})$ be any other block (e.g., chat-template tokens and/or future tokens). For any query $\boldsymbol{q}$,

$$\text{Attn}\left(\boldsymbol{q}; \begin{bmatrix} \boldsymbol{K} \\ \boldsymbol{K}_{\text{fixed}} \end{bmatrix}, \begin{bmatrix} \boldsymbol{V} \\ \boldsymbol{V}_{\text{fixed}} \end{bmatrix}\right) = \frac{\text{Mass}(\boldsymbol{q}; \boldsymbol{K})}{\text{Mass}(\boldsymbol{q}; \boldsymbol{K}) + \text{Mass}(\boldsymbol{q}; \boldsymbol{K}_{\text{fixed}})} \, \text{Attn}(\boldsymbol{q}; \boldsymbol{K}, \boldsymbol{V})$$
$$+ \frac{\text{Mass}(\boldsymbol{q}; \boldsymbol{K}_{\text{fixed}})}{\text{Mass}(\boldsymbol{q}; \boldsymbol{K}) + \text{Mass}(\boldsymbol{q}; \boldsymbol{K}_{\text{fixed}})} \, \text{Attn}(\boldsymbol{q}; \boldsymbol{K}_{\text{fixed}}, \boldsymbol{V}_{\text{fixed}}). \tag{5}$$

Now suppose we replace $(\boldsymbol{K}, \boldsymbol{V})$ by compact parameters $(\boldsymbol{C}_k, \boldsymbol{\beta}, \boldsymbol{C}_v)$. If for queries of interest we ensure

$$\text{Attn}(\boldsymbol{q}; \boldsymbol{K}, \boldsymbol{V}) \approx \text{Attn}(\boldsymbol{q}; \boldsymbol{C}_k, \boldsymbol{\beta}, \boldsymbol{C}_v) \quad \text{and} \quad \text{Mass}(\boldsymbol{q}; \boldsymbol{K}) \approx \text{Mass}(\boldsymbol{q}; \boldsymbol{C}_k, \boldsymbol{\beta}),$$

then the mixture identity (5) implies that the full attention output is preserved even when *arbitrary* $\boldsymbol{K}_{\text{fixed}}, \boldsymbol{V}_{\text{fixed}}$ are appended later, because both mixture weights and the compacted-block contribution match. This means we can use objectives (1)–(2) to do one-shot compaction without knowing future keys/values.

**Why biases matter.** If $C_k$ is chosen as a subset of $K$ and we do not use attention biases, then $\mathrm{Mass}(q; C_k) \leq \mathrm{Mass}(q; K)$ for every $q$, so the compacted block systematically receives too little global weight in (5). A bias $\beta$ introduces multiplicative weights $\exp(\beta_j)$ so that each retained key can represent the mass of many removed keys, making mass matching feasible. Furthermore, even if $C_k$ is not chosen as a subset of $K$, there are cases where the bias is necessary to match attention mass, such as $q = 0$, which requires matching $T$ vs $t$. For these reasons, introducing attention biases after compaction is very natural.

**Stable computation.** In implementation, we evaluate $\mathrm{Mass}$ and $\mathrm{Attn}$ using the standard per-query max-shift for numerical stability. For logits $\ell$ with $s = \max_j \ell_j$, we compute $\sum_j \exp(\ell_j) = \exp(s) \sum_j \exp(\ell_j - s)$ and similarly for the numerator of $\mathrm{Attn}$. This does not change any of the identities above. For OMP key selection, we operate on shifted features $\exp(\ell - s)$ and shifted targets $\sum_j \exp(\ell - s)$.

# B. Experimental Details

### B.1. Models

We evaluate `Qwen3-4B` (Qwen Team, 2025), `Llama-3.1-8B-Instruct` (Llama Team, 2024), and `Gemma-3-12b-it` (Gemma Team, 2025). We use a maximum generation length of 2048 tokens and each model's default decoding settings (temperature, top-$k$, and top-$p$). For long-context benchmarks (LongHealth), we use the `Qwen3-4B-Instruct-2507` variant instead of `Qwen3-4B` because it has a longer native sequence length.

### B.2. Datasets

Our default evaluation dataset is **QuALITY** (Pang et al., 2022), a question-answering dataset testing comprehension of 5-8k-token passages. For our evaluations, we first compact the article context and then evaluate performance on the resulting QA task by batched decoding over the associated questions. We believe that pairing each article with many comprehension-focused multiple-choice questions makes QuALITY a controlled and informative benchmark for compaction.

We evaluate performance over the first 50 articles in the validation set (894 questions). Due to the computational cost of Cartridges (∼5 hours per context), we only evaluate it on Qwen3-4B, on the first 20 articles (357 questions). Original-cache accuracy on this subset (75.0%) exceeds that of the full set (71.1%).

We evaluate chunked compaction on **LongHealth** (Adams et al., 2024) by concatenating 5 patient records per context. This yields 4 contexts of roughly 60k tokens each, with 100 multiple-choice questions per context. We compact each context in 5 chunks before evaluating QA performance. While Eyuboglu et al. (2025) concatenate 10 patient records, we found that our evaluated base models' performance degraded beyond ∼100k tokens, which we attribute to limitations in long-context extrapolation of the model rather than the compaction procedure. In contrast, Cartridges optimizes a single latent representation across chunks and is thus not sensitive to long-context brittleness in the base model.

We show an example context snippet and question below for QuALITY and LongHealth. These tasks can involve heavy information-extraction and reasoning across the long context, making them an ideal and difficult testbed for compaction.

---

**`QuALITY`**

**Context (excerpt):**
THE GIRL IN HIS MIND By ROBERT F. YOUNG
Transcriber's Note: This etext was produced from Worlds of Tomorrow April 1963
Extensive research did not uncover any evidence that the U.S. copyright on this publication was renewed.
Every man's mind is a universe with countless places in which he can hide—even from himself!
... (5-8k token article)

**Sample Question:**
Why does Deirdre get so upset when Blake Past suggests she go to prom with the young man?
A) Because Blake is trying to guilt Deirdre into going with the young man by telling her that it'll ease her conscience.
B) Because Deirdre has fallen in love with Blake, despite his age, and wants him to take her to the prom.
C) Because Blake is acting like he's her father, which is a sensitive topic for Deirdre because she lost her real parents.
D) Because the young man gave up his right arm in order to afford tickets to the prom, and this disgusts Deirdre.

---

```
LongHealth
```

**Context (excerpt):**
Dear colleague,
We wish to provide an update regarding Mrs. Anna Sample, born on 01.01.1970. She was admitted to our clinic from 01/01/2017 to 01/02/2017.
Diagnosis: Diffuse large B-cell lymphoma of germinal center type; ID 01/2017
- Ann-Arbor: Stage IV
- R-IPI: 2 (LDH, stage)
- CNS-IPI: 2
- Histology: Aggressive B-NHL (DLBCL, NOS); no evidence of t(14;18) translocation. Ki-67 at 40%. Positive reaction to MUM1, numerous CD68-positive macrophages. Negative reaction to ALK1 and TdT.
- cMRI: Chronic inflammatory lesions suggestive of Multiple Sclerosis (MS)
- CSF: no evidence of malignancy
- Bone marrow aspiration: no infiltration from the pre-existing lymphoma.
Current treatment:
Initiated R-Pola-CHP regimen q21
- Polatuzumab vedotin: 1.8mg/kg on Day 1.
... (10-12k tokens per patient; 5 patients)

**Sample Question:**
Mrs. Sample received multiple radiologic examinations. In which order did she receive them?
A) MR spine > MR Head > CT Thoracic Spine > CT Whole Body
B) MR Brain > MR thoracic/lumbar spine > CT chest/abdomen/pelvis > CT Thoracic Spine > CT Thoracic Spine > CT chest/abdomen/pelvis > Abdominal ultrasound > MR Spine > Whole-body PET/CT > Liver MRI
C) MR cervical spine > MR Head > CT Whole Body > CT Thoracic Spine
D) MR Brain > CT Thoracic Spine > MR cervical spine > CT Whole Body
E) CT Thoracic Spine > CT Whole Body > MR cervical spine > MR Head

**Additional benchmarks.** To test generalization beyond multiple-choice comprehension, we additionally evaluate on three benchmarks spanning generative QA, retrieval-intensive tasks, and diverse real-world domains. We follow the same one-shot protocol as above (compact the context once, then evaluate all associated queries), keeping chat-template tokens fixed.

**QASPER** (Dasigi et al., 2021) is a free-form question-answering dataset over NLP research papers, where answers are short generated spans rather than multiple-choice options. We report token-level F1 against the reference answers over 562 paper contexts and 2,010 questions, evaluating Qwen3-4B and Llama-3.1-8B-Instruct. This tests whether compaction preserves precise extractable information under generative decoding, not just answer selection.

**RULER** (Hsieh et al., 2024) is a synthetic suite of needle-in-a-haystack, multi-hop tracing, and aggregation tasks that stress long-context retrieval. We evaluate over the 6,500 RULER tasks packaged in `kvpress` (Devoto et al., 2025) on Qwen3-4B and Llama-3.1-8B-Instruct. For Llama-3.1-8B-Instruct we additionally report numbers for AdaKV, SnapKV, Finch, KeyDiff, TOVA, and PyramidKV taken directly from the public `kvpress` leaderboard, alongside our own runs of KVzip (the prior state of the art on this benchmark). For RULER we use only the `AM-HighestAttnKeys-fast` variant (repeat-prefill queries, no self-study or OMP), so these numbers reflect our fastest, not our strongest, configuration.

**LongBench v2** (Bai et al., 2025) covers code repositories, tables, multilingual documents, and long structured contexts. Because the base models we use achieve low absolute accuracy on the full benchmark (making method differences hard to resolve), we first run the full-context and no-context baselines on the full set, then restrict evaluation to the 47 contexts (max 97k tokens; average 42k tokens) that the model answers correctly with the full context but incorrectly without it. This concentrates the evaluation on questions where the context is decisive so differences between methods are easier to resolve; we re-evaluate every method (including `no ctx` and `original`) on this subset, so all reported numbers are directly comparable within the table. We evaluate Qwen3-4B-Instruct-2507 on this subset.

## B.3. Baselines

**Cartridges.** We run experiments using the Cartridges repository (Eyuboglu et al., 2025). For each QuALITY article, we synthesize 32,768 self-study samples (25–30M tokens) using the default settings and train for 1 epoch with the default learning rate ($2 \times 10^{-2}$). We replicate the same order of magnitude of training compute for a fair comparison (Table 5). We use the same evaluation setup as in our other experiments. Optimizing training or using more diverse self-study samples may greatly speed up results; we use the default settings provided in the open-source repository.

| Dataset | Context | Self-Study Tokens | H100-hours |
|---|---|---|---|
| LongHealth-5 | 60k | 100–150M | $\sim 15$ |
| Quality | 7k | 25–30M | $\sim 5$ |
| LongHealth-10 (Original) | 120k | 300M | - |

*Table 5.* **Training compute for Cartridges.** We sample approximately the same number of self-study tokens per context token as in the original Cartridges implementation.

**H2O, SnapKV, PyramidKV, and KVzip.** We evaluate H2O, SnapKV, PyramidKV, and KVzip (Zhang et al., 2023; Li et al., 2024; Cai et al., 2025; Kim et al., 2025b). These methods select a subset of keys based on highest attention scores under a chosen set of queries.

H2O was originally proposed as an online eviction policy; here we use its one-shot compaction analogue, which we term **H2O+**. Concretely, during context prefill we collect all queries for each KV-head position and then select the top-$k$ tokens per head by attention score, rather than evicting tokens sequentially based on only partial query sets. Moreover, because we compact only the content portion of the context—keeping chat-template tokens and the question portion fixed—we do not explicitly retain all recent tokens as in the original paper. Finally, we use the GQA variant of this method (and of the methods below): instead of duplicating KV heads before compaction (as some implementations do), we aggregate queries from all query heads that attend to a given KV head into a shared query set for that KV head.

SnapKV and PyramidKV are question-aware in their original form (they compact after observing the downstream question). To make them question-agnostic and comparable to our setting, we instead generate a single mock question via self-study and use it to construct the query set used for key scoring. We use max-pooling with a kernel-size of 7, consistent with original hyperparameters.

**KVMerger.** KVMerger (Wang et al., 2024) is a token-*merging* method rather than a token-eviction method: it groups consecutive keys whose cosine similarity exceeds a threshold into clusters, then merges each cluster into a single representative KV pair via a Gaussian-kernel weighted average of its values (anchored at the highest-attention token in the cluster). This contrasts with Attention Matching, which globally subsets keys across the entire prefix and then refits the retained values and per-token biases to approximate the attention contribution of the *unchosen* keys, rather than locally averaging adjacent tokens. We use the authors' default similarity threshold and Gaussian kernel width, and apply the same one-shot, question-agnostic protocol (repeat-prefill query set, fixed chat-template tokens) used for the other baselines so that the reported compaction ratio is comparable.

Table 6 summarizes the design choices for each baseline.

**DuoAttention.** For Llama-3.1-8B-Instruct, we additionally attempted to compare against DuoAttention (Xiao et al., 2025), which supports this model. We implement DuoAttention by partitioning heads into *streaming* and *retrieval* heads: for streaming heads we evict the article portion of the KV cache, while for retrieval heads we retain the full KV cache. The overall compaction ratio determines the fraction of heads assigned to each category. However, at the compaction ratios we evaluated (retaining up to $0.2\times$ the original cache), DuoAttention performed poorly, so we omit it from the main figure.

**Summarization.** As a baseline, we summarize the article text and use the summary in place of the original article for downstream evaluation. We evaluate several prompt variants shown in Table 7.

| Method | Queries | Key selection | $\beta$ | Values | Nonuniform |
|---|---|---|---|---|---|
| H2O+ | context-prefill | HighestAttnKeys | none | direct | none |
| SnapKV (agnostic) | mock-question | HighestAttnKeys | none | direct | none |
| PyramidKV (agnostic) | mock-question | HighestAttnKeys | none | direct | linearly decreasing |
| KVzip-uniform | repeat-prefill | HighestAttnKeys | none | direct | none |
| KVzip | repeat-prefill | GlobalHighestAttnKeys | none | direct | global top-$k$ |
| KVMerger | none | cosine-similarity merge | none | merged | none |

*Table 6.* **Baseline configurations.** "Mock-question" denotes a single synthetic question generated via self-study to instantiate originally question-aware methods in a question-agnostic setting. "Direct" values indicate that retained keys keep their original values (i.e., no value fitting with least squares); "merged" values denote Gaussian-kernel weighted averaging of clustered tokens (KVMerger).

| Name | Prompt |
|---|---|
| summarize | Summarize the following text:
{article_text}
Summary: |
| summarize_indepth | Summarize the following text in depth:
{article_text}
Summary: |
| summarize_keypoints_questions | Summarize the following text, providing all key points that might be necessary to answer comprehension questions:
{article_text}
Summary: |
| summarize_concise | Summarize the following text concisely:
{article_text}
Summary: |
| summarize_very_concise | Summarize the following text very concisely:
{article_text}
Summary: |

*Table 7.* **Summarization prompts.**

# C. Algorithmic Implementation Details

We implement each of our per-head algorithms sequentially, iterating through each KV-head in the model. We did not find major speedups through batching computation across heads.

## C.1. OMP Speedups

The periodic-refit OMP key selection algorithm is shown in Algorithm 2. It introduces two hyperparameters, $k$ and $\tau$: $k$ is the number of keys selected per greedy iteration, and $\tau$ is the number of iterations between NNLS refits (i.e., we recompute weights once every $k\tau$ newly added keys). Our main algorithm implicitly uses $k = 1, \tau = 1$, and we use $k = 4$ and $\tau = 2$ in our fast variant.

---

**Algorithm 2** OMP Key Selection (Periodic Refit)

---

**Require:** Original keys $\boldsymbol{K} \in \mathbb{R}^{T \times d}$, queries $\boldsymbol{Q} \in \mathbb{R}^{n \times d}$, budget $t$, top-$k$ $k$, NNLS interval $\tau$
**Ensure:** Indices $S$, weights $\mathbf{w}$ (where $\boldsymbol{\beta} = \log \mathbf{w}$)

1: $\boldsymbol{\Phi}_{ij} \leftarrow \exp\left(\frac{1}{\sqrt{d}} \boldsymbol{q}_i \boldsymbol{K}_j^\top\right)$ {Mass feature matrix}
2: $\mathbf{m}_i \leftarrow \sum_{j=1}^{T} \boldsymbol{\Phi}_{ij}$ {Target mass vector}
3: $S \leftarrow \emptyset, \quad \mathbf{w} \leftarrow [\,], \quad \mathbf{r} \leftarrow \mathbf{m}$
4: **for** $u = 1$ **to** $t$ **do**
5: $\quad c_j \leftarrow \mathbf{r}^\top \boldsymbol{\Phi}_{:j} \quad \forall j \notin S$ {Correlation scores}
6: $\quad J^\star \leftarrow \text{TopK}\left(\{c_j\}_{j \notin S}, k\right)$ {Select top-$k$ new keys}
7: $\quad S \leftarrow S \cup J^\star$
8: $\quad$ **if** $u \bmod \tau = 0$ **or** $u = t$ **then**
9: $\quad\quad \mathbf{w} \leftarrow \arg\min_{\mathbf{w} \geq 0} \|\boldsymbol{\Phi}_{:S}\mathbf{w} - \mathbf{m}\|_2^2$ {Solve NNLS}
10: $\quad\quad \mathbf{r} \leftarrow \mathbf{m} - \boldsymbol{\Phi}_{:S}\mathbf{w}$ {Update residual}
11: $\quad$ **end if**
12: **end for**
13: **return** $S, \mathbf{w}$

---

## C.2. Linear Algebra Subroutines

**Numerics.** We compute $\boldsymbol{C}_k, \boldsymbol{\beta}$, and $\boldsymbol{C}_v$ in FP32, then cast to BF16 for storage and subsequent use.

**Least Squares.** We evaluated three solvers for the least-squares problems in Eq. 3: `torch.linalg.lstsq`, `torch.linalg.pinv`, and a Cholesky-based approach using `torch.linalg.cholesky` with `torch.cholesky_solve`. Concretely, we compute

1. `torch.linalg.lstsq`: We directly compute the solution to the optimization problem of $\arg\min_M \|XM - Y\|_F^2$, as $M = \text{torch.linalg.lstsq}(X, Y)$. On CUDA, PyTorch uses the `gels` driver, performing QR decomposition on $X = QR$, transforming the target $Y$ by left multiplication with $C = Q^T$, then solves the triangular problem of $RM = C$.

2. `torch.linalg.pinv`: We compute the Moore-Penrose pseudo-inverse explicitly and then right multiply by $Y$, i.e. $M = \text{torch.linalg.pinv}(X)Y$.

3. `torch.linalg.cholesky`: We solve the normal equations $X^T X M = X^T Y$ by computing the Cholesky decomposition of the symmetric positive-definite matrix $A = X^T X$:

$$LL^T = X^T X$$

where $L = \text{torch.linalg.cholesky}(A)$. The solution is obtained by solving the two-step triangular system $LZ = X^T Y$ and $L^T M = Z$ via `torch.cholesky_solve`.

Across our experiments, solution quality ranked `lstsq` > `cholesky` > `pinv`, though the difference was minimal. In terms of runtime, `cholesky` was fastest, followed by `lstsq`, then `pinv`. We therefore use `torch.linalg.lstsq`

in all experiments. We also tested $\ell_2$ regularization when estimating $\boldsymbol{\beta}$ and $\boldsymbol{C}_v$ by computing the regularized solution $(X^T X + \lambda I)^{-1}$, but found that it degraded performance across all positive values of $\lambda$.

**Nonnegative Least Squares (NNLS).**  We implement an NNLS solver with projected gradient descent. When `iters=0`, we compute an unconstrained least-squares solution with `torch.linalg.lstsq` and then clamp to enforce $B \geq \epsilon$ (and optionally $B \leq u$). For `iters>0`, we warm-start from this clamped solution and run projected gradient descent for `iters` steps on $\frac{1}{2}\|MB - y\|_2^2$, using a fixed step size $1/L$ where $L \approx \|M\|_2^2$ is estimated via a few power-iteration steps.

Algorithm 3 describes the NNLS solver used throughout. We initialize from an unconstrained least-squares solution followed by clamping, and (when `iters>0`) refine the solution using projected gradient descent on $\frac{1}{2}\|MB - y\|_2^2$ with a fixed step size.

---

**Algorithm 3** NNLS via Projected Gradient Descent

---

**Require:** Matrix $M \in \mathbb{R}^{n \times p}$, target $y \in \mathbb{R}^n$, iterations `iters`, lower bound $\epsilon \geq 0$, optional upper bound $u$
**Ensure:** Nonnegative weights $B \in \mathbb{R}^p$
 1: $B^{(0)} \leftarrow \arg\min_B \|MB - y\|_2^2$ {Unconstrained least squares}
 2: $B^{(0)} \leftarrow \max(B^{(0)}, \epsilon)$ {Clamp to enforce $B \geq \epsilon$}
 3: **if** upper bound $u$ is specified **then**
 4:     $B^{(0)} \leftarrow \min(B^{(0)}, u)$
 5: **end if**
 6: **if** `iters` $= 0$ **then**
 7:     **return** $B^{(0)}$
 8: **end if**
 9: Estimate $L \approx \|M\|_2^2$ via power iteration
10: $\eta \leftarrow 1/L$ {Fixed step size}
11: **for** $s = 0$ **to** `iters` $- 1$ **do**
12:     $g^{(s)} \leftarrow M^\top (MB^{(s)} - y)$ {Gradient}
13:     $\tilde{B} \leftarrow B^{(s)} - \eta g^{(s)}$ {Gradient step}
14:     $B^{(s+1)} \leftarrow \max(\tilde{B}, \epsilon)$ {Projection}
15:     **if** upper bound $u$ is specified **then**
16:         $B^{(s+1)} \leftarrow \min(B^{(s+1)}, u)$
17:     **end if**
18: **end for**
19: **return** $B^{(\text{iters})}$

---

For OMP Keys, we set $\text{iters} = 0$ as we observed that the unconstrained solution rarely yields negative weights. For Highest Attention Keys, we use `iters=2`.

**Stabilizing $\boldsymbol{\beta}$.**  Our pipeline first fits $\boldsymbol{\beta}$ to match attention mass and then fits $\boldsymbol{C}_v$ to match attention outputs. A potential failure mode of this two-stage procedure is that mass matching can assign extremely small weights to some selected keys (i.e., very negative $\boldsymbol{\beta}$, or effectively $\boldsymbol{\beta} = -\infty$). Such keys may have little effect on the mass objective yet still be useful for reducing attention-output error; however, once $\boldsymbol{\beta}$ is very negative, the corresponding key cannot contribute to the attention output, regardless of $\boldsymbol{C}_v$.

To mitigate this, we apply simple stability constraints on $\boldsymbol{\beta}$. For *Highest Attention Keys*, we replace NNLS with a bounded least-squares over $w = \exp(\boldsymbol{\beta})$, enforcing $e^{-3} \leq w_j \leq e^3$ (equivalently, $\boldsymbol{\beta}_j \in [-3, 3]$). The projected gradient method above supports these box constraints directly.

For *OMP Keys*, which rarely produces extreme $\boldsymbol{\beta}$ values, we instead use a lightweight pruning rule: after greedy selection, we discard any selected keys with $\boldsymbol{\beta} < -7$ and continue selection until all retained keys satisfy $\boldsymbol{\beta} \geq -7$. We also cap large biases during weight fitting by enforcing $w_j \leq e^7$ (equivalently, $\boldsymbol{\beta}_j \leq 7$). In practice, these OMP safeguards did not yield noticeable performance gains, since OMP rarely assigns very low or high $\boldsymbol{\beta}$ values; we nonetheless include them as simple, theoretically sound stabilizers.

## C.3. Chunked Compaction Details

Here we provide implementation details for chunked compaction. Given a document split into $N$ contiguous chunks, we apply per-layer/per-head compaction independently to each chunk and concatenate the resulting compacted KV segments to form the final cache:

$$[\text{prefix}] \; + \; [\text{compacted chunk}_1] \; + \; \cdots \; + \; [\text{compacted chunk}_N] \; + \; [\text{suffix}],$$

where the prefix and suffix are kept fixed with $\boldsymbol{\beta} = 0$, and each chunk contributes learned $(\boldsymbol{C}_k, \boldsymbol{\beta}, \boldsymbol{C}_v)$.

We consider two variants: KV-based and text-based chunking.

**KV-based chunking** prefills the full context once, slices out each chunk's KV states, compacts those tensors, and concatenates the compacted chunks. To generate self-study reference queries for chunk $i$, we construct a cache by concatenating the corresponding KV tensors:

$$[\text{prefix}] \; + \; [\text{chunk}_i] \; + \; [\text{suffix}],$$

leaving any sliding-window layers unchanged (copied from the full prefill). When prefilling self-study conversations to extract reference queries, we continue positional indexing from the original sequence length so that appended tokens receive the same RoPE phases as in the uncompacted run. These choices preserve the global positional semantics of the original sequence: although the physical cache size is smaller, its logical length and RoPE phases match those of the uncompacted run.

**Text-based chunking** prefills and compacts each chunk in isolation as $[\text{local prefix}] + [\text{local chunk}_i] + [\text{local suffix}]$, using chunk-local positions starting at $0$, and then applies a uniform RoPE phase shift to the compacted keys to align them to the chunk's original global offset in the full context (i.e., a rotation by $\Delta = p_{\text{global}} - p_{\text{local}}$). For models with sliding-window attention (e.g., Gemma), we handle sliding layers by instantiating the final cache's sliding-window KV from the last chunk (including the global suffix) with the same positional shift applied.

## C.4. Self-study

In our implementation of self-study, we run two "instances" of the model:

1. **Conversation-starter generation.** Instance A is prompted with $\mathcal{C}$ and a seed prompt (e.g., "Write 3 different questions separated by newlines that test understanding of the context."). We sample

   $$a \sim \text{LM}(\cdot \mid \mathcal{C}, \text{seed}),$$

   then post-process $a$ into $m$ separate conversation starters $(a_i)_{i=1}^m$.

2. **Response generation.** For each $a_i$, we sample from instance $B$ a response

   $$b_i \sim \text{LM}(\cdot \mid \mathcal{C}, a_i).$$

   During instance B's prefill on $a_i$ and decoding of $b_i$, we record, for each attention head, all query vectors used by the model.

We include a set of fixed prompts (e.g., "Aggregate all key facts mentioned in the context.") by directly setting $a$ to the prompt string and sampling the corresponding response from instance B.

We use vLLM (Kwon et al., 2023) to generate self-study conversations, and then we prefill these conversations with a forward hook that extracts query vectors at each KV-head. To control the number of extracted queries, we apply reservoir sampling and cap the buffer at 50,000 query vectors per KV-head. In practice, our self-study configuration typically yields fewer than this cap (avg. $\sim 16{,}000$ per head for QuALITY).

For on-policy self-study and repeat-prefill, we first generate each self-study sequence and prefill the first layer to extract its queries. Then, for each layer $L$, we prefill layer $L$ using the compacted cache for layers $0$ through $L-1$, and extract the query vectors at layer $L$.

| Name | Prompt |
|---|---|
| 3-question | Write 3 questions that test understanding of different parts of the context. Answer with just the 3 questions and options (do not say the correct answer), each one separated with 2 newlines. |
| summarize | Summarize the main points of the context. |
| structure_json | Structure the information in JSON form and include all important details like dates, times, names, and numerical values. |
| aggregate | Aggregate all the key facts mentioned in the context. |

*Table 8.* **Prompt templates used in self-study.**

**Prompts.**  We use the prompt templates in Table 8.

We obtain one conversation starter directly from each of summarize, structure_json, and aggregate. For 3-question, we sample a response from instance A and split the generated output into three conversations starters. We then sample one response per starter.

## D. Impact of Reference Queries

We present uniform compaction results under different reference query generation strategies in Figure 6. We evaluate log(Perplexity) of the original cache's answers on 20 QuALITY instances (357 questions). We plot this rather than downstream accuracy because it has lower variance and correlates well with downstream accuracy.

For random vectors, we sample IID from a normal distribution with mean 0 and standard deviation 1, and then apply $q$-norm scaling for the respective attention heads. We omit this step in random-vectors-no-qnorm. In ss-5k and ss-plus-repeat-5k, we subsample 5,000 query vectors uniformly at random after running the corresponding query generation method.

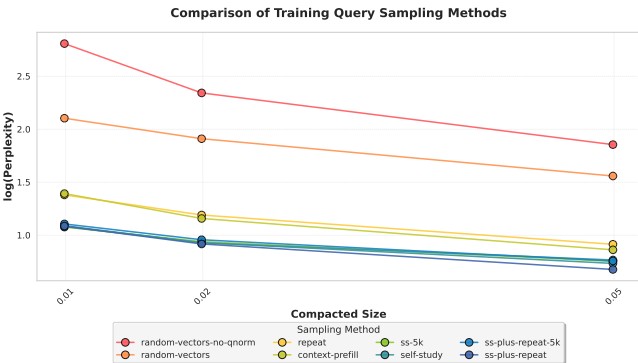

*Figure 6.* **Reference query sampling comparison.** We compare eight variants for sampling queries. Self-study-based methods perform best, especially at the greatest compaction ratios, with repeat and context-prefill close behind. Subsampling reference queries preserves performance.

# E. Head Budgets

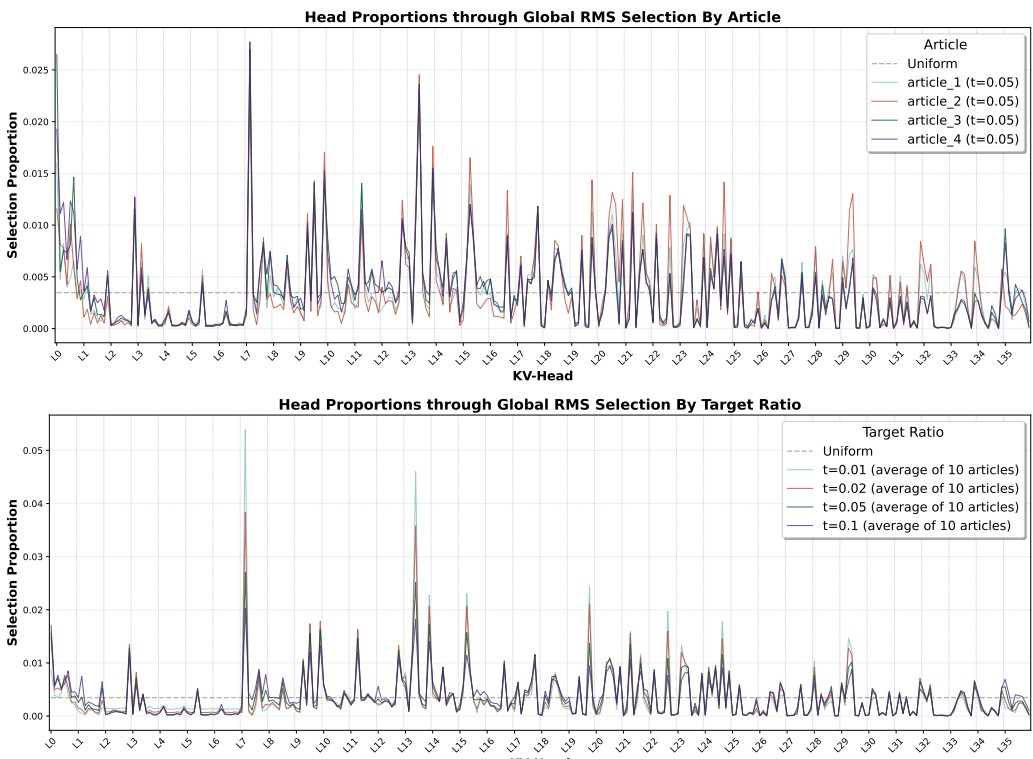

*Figure 7.* **Head proportions induced by global highest-attention selection.** We plot the fraction of total selected keys attributed to each of the 288 KV heads (36 layers, 8 heads per layer) in Qwen3-4B. **Top:** proportions across 10 QuALITY articles at a fixed global compaction ratio of 0.05. **Bottom:** average proportions across multiple global compaction ratios. The low variance across articles and target ratios indicates stable head importance patterns.

This appendix provides additional evidence motivating nonuniform KV compaction and visualizes the per-head budgets learned by our method.

**Global key selection induces stable head proportions.** KVzip (Kim et al., 2025b) selects keys based on highest attention scores *globally* across all KV heads, without enforcing per-head quotas. Under a fixed global top-$k$ budget, this induces an implicit allocation in which different heads retain different fractions of the total KV capacity.

Figure 7 shows the implicit head proportions induced by global highest-attention selection in Qwen3-4B, supporting the use of a fixed, precomputed nonuniform schedule.

**Head sensitivity curves.** Following Section 3.4, we directly measure head sensitivity by varying the compaction ratio of individual heads while holding all other heads fixed. Figure 3 in the main paper shows representative sensitivity curves for Qwen3-4B.

To demonstrate that this behavior generalizes across model families, Figure 8 shows analogous head sensitivity curves for Llama-3.1-8B-Instruct. In both models, we observe substantial differences across heads: some heads are largely insensitive to KV capacity, while others benefit significantly from retaining additional keys.

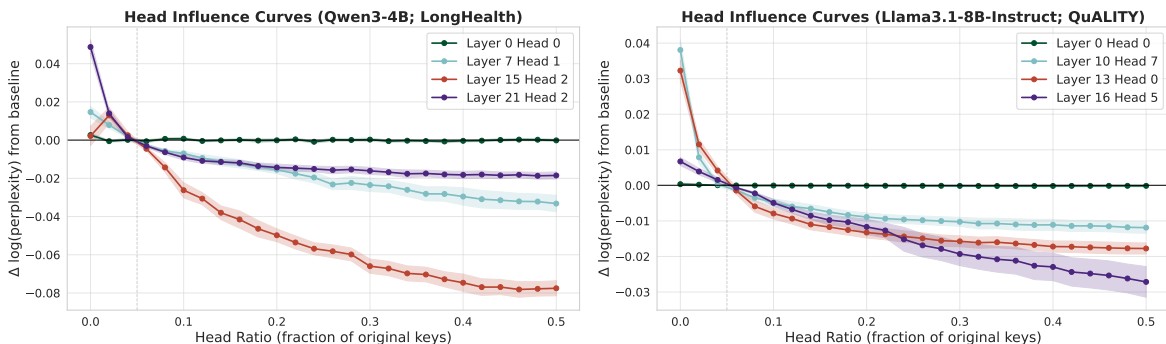

*Figure 8.* **More Head sensitivity curves.** Analogous to Figure 3, showing loss change as a function of per-head compaction ratio for four selected heads. **Left:** Qwen3-4B with LongHealth exhibits similar patterns to Figure 3. **Right:** Llama-3.1-8B-Instruct also shows head-dependent sensitivity patterns: some heads are largely insensitive to KV capacity, while others benefit substantially from retaining more keys.

**Optimized head budgets.**    Using these sensitivity curves, we apply the budget-allocation procedure described in Section 3.4 and Algorithm 4 to derive model-specific nonuniform head budgets. Figure 9 compares the resulting optimized head proportions to the average head proportions induced by global highest-attention key selection.

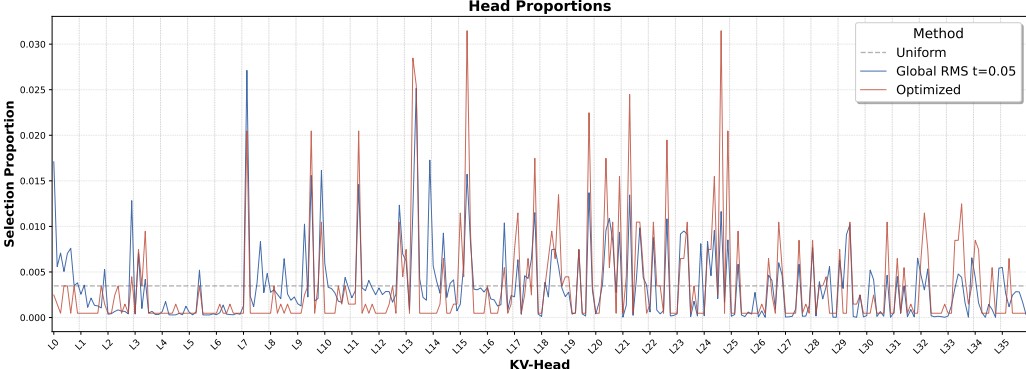

*Figure 9.* **Optimized head budgets.** We compare the head proportions learned by our nonuniform allocation algorithm to those induced by global highest-attention key selection. The optimized schedule has similar structure as the global highest attention scores schedule. The optimized schedule tends to assign more budget to later layers and less to earlier layers, contrary to PyramidKV (Cai et al., 2025).

**Algorithm 4** Nonuniform KV Allocation via Greedy Swaps

**Require:** Heads $h = 1, \ldots, H$; step size $\eta > 0$ (share units); per-head sensitivity curves $J_h(\rho)$ mapping compaction ratio $\rho \in [0, 1]$ to loss; reference overall compaction ratio $r_0$ (we use $r_0 = 0.05$).
**Ensure:** Shares $\{p_h\}$ with $\sum_h p_h = 1$ and $p_h \geq 0$.

1: **Initialize.** $p_h \leftarrow \frac{1}{H}$ for all $h$.
2: **repeat**
3:     **Map shares to ratios.** $\rho_h \leftarrow p_h \cdot H \cdot r_0$ for all $h$.
4:     **Compute marginals.** For each head $h$,

$$g_h^+ \leftarrow \begin{cases} J_h(\rho_h) - J_h(\rho_h + \eta'), & \text{if } \rho_h + \eta' \leq 1, \\ -\infty, & \text{otherwise,} \end{cases} \qquad g_h^- \leftarrow \begin{cases} J_h(\rho_h - \eta') - J_h(\rho_h), & \text{if } \rho_h \geq \eta', \\ +\infty, & \text{otherwise,} \end{cases}$$

    where $\eta' = \eta \cdot H \cdot r_0$ is the step size in ratio space.
5:     **Select best swap.** $b^\star \leftarrow \arg\max_h g_h^+, \quad a^\star \leftarrow \arg\min_h g_h^- \quad$ with $a^\star \neq b^\star$.
6:     **if** $g_{b^\star}^+ > g_{a^\star}^-$ **then**
7:         $p_{a^\star} \leftarrow p_{a^\star} - \eta; \quad p_{b^\star} \leftarrow p_{b^\star} + \eta$.
8:     **else**
9:         **break** {No improving swap remains.}
10:     **end if**
11: **until** termination
12: **return** $\{p_h\}$.

# F. Further Results

### F.1. Mean vs. RMS vs. Max Aggregation

Several prior methods—including H2O, SnapKV, PyramidKV, KVzip, and our *AM-HighestAttnKeys*—select a subset of keys based on attention scores under a collection of query vectors. Concretely, for each key $K_j$ and reference query $q_i$, we compute the attention weight

$$a_{i,j} = \text{softmax}(q_i K^\top)_j.$$

To rank keys globally, these per-query attention weights must be aggregated across queries into a single importance score per key, after which the top-$t$ keys are retained.

We consider three aggregation functions:

$$s_j^{\text{mean}} = \frac{1}{n} \sum_{i=1}^n a_{i,j}, \quad s_j^{\text{rms}} = \sqrt{\frac{1}{n} \sum_{i=1}^n a_{i,j}^2}, \quad s_j^{\text{max}} = \max_i a_{i,j}.$$

Figure 10 compares downstream performance across these choices for four baseline methods as well as our Highest Attention Keys approach. There is no choice that is consistently best across methods, though RMS offers a good balance and we adopt it in our method.

### F.2. Reconstruction vs. Downstream Accuracy

In addition to QA accuracy, we also compute loss (log-perplexity) on the *original-cache generations*: we evaluate the negative log-likelihood of responses sampled from the full cache, but conditioned on the compacted cache. This metric measures how well a compacted cache preserves the model's token-level behavior, differing from the forward KL divergence only by a constant additive term. Figure 11 (left) replicates our main trade-off plot using this loss metric, and Figure 11 (right) compares downstream multiple-choice accuracy directly against log-perplexity across methods and compaction ratios.

Across attention-matching methods and token-selection baselines, we observe that points roughly interpolate between the *no-context* results (high loss, low accuracy) and the *full-context* results (low loss, high accuracy). As expected, improvements in reconstruction tend to translate into improvements in QA accuracy.

**log(Perplexity) vs Compacted Size by Aggregation Method**

*Figure 10.* **Effect of aggregation function on key selection.** We plot accuracy versus compaction ratio for four baseline methods and our Highest Attention Keys method on a subset of 20 QuALITY articles using Qwen3-4B. All methods select the top-$t$ keys based on aggregated attention scores over a shared set of reference queries.

Two notable deviations from this trend are **Cartridges** and **summarization**. Cartridges tends to achieve slightly *lower* reconstruction loss than methods with similar downstream accuracy, consistent with its objective: it performs end-to-end latent-cache optimization using a distillation-style loss that targets the original model's behavior, directly optimizing token-level fidelity. In contrast, summarization often yields *higher* reconstruction loss than methods with similar accuracy: it discards substantial token-level detail but can preserve (or even emphasize) the subset of information most useful for answering comprehension questions. In other words, summarization can be a good task heuristic despite being a poor reconstruction mechanism.

Overall, these results support two conclusions: (i) reconstruction loss is a useful proxy for comparing compaction methods *within* families that aim to preserve attention behavior, and (ii) it should not be treated as a universal predictor of downstream performance when methods fundamentally change the objective. We further note that self-log-perplexity in long-context evaluations can be skewed on many language models due to entropy blowup as generations grow longer (Braverman et al., 2020; Cao et al., 2025). The perplexity numbers reported here are for Qwen3-4B on QuALITY (contexts $< 10$k tokens), where we did not observe such entropy blowup, though we did observe this at longer context lengths.

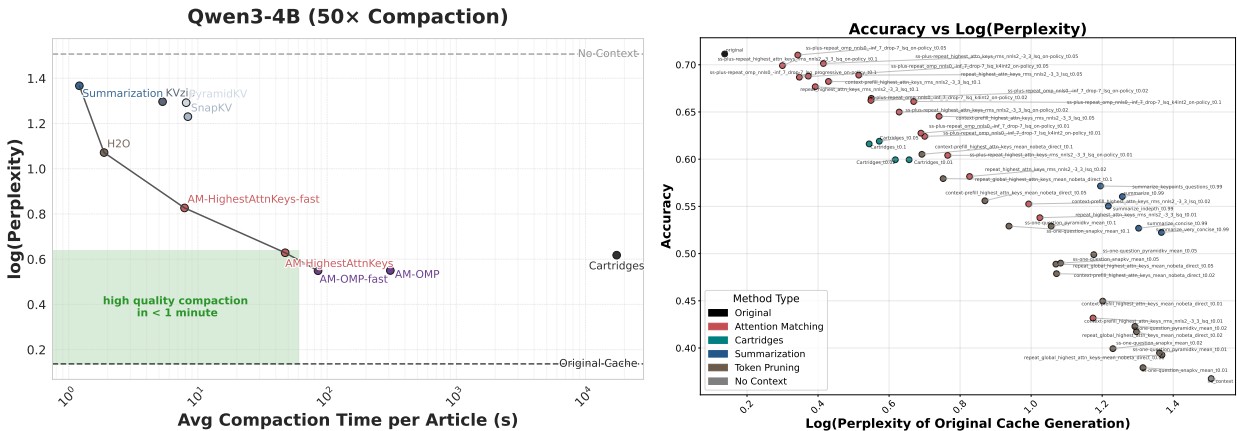

*Figure 11.* **Reconstruction loss vs. downstream accuracy. Left:** Loss-based analogue of Figure 1, instead reporting log-perplexity on original-cache generations. **Right:** Downstream multiple-choice accuracy plotted against this loss across various methods and compaction ratios.

## F.3. Extension: Online Compaction

For long-horizon reasoning, we may want to compact *mid-trajectory* so the model can continue decoding after producing thousands of intermediate tokens.

| Phys Len | Eff Len | AIME (/30) | Notes |
|---|---|---|---|
| 2048 | 2048 | 1.25 | |
| 4096 | 4096 | 7.75 | |
| 8192 | 8192 | 13 | |
| *With online compaction* | | | |
| 2048 | 4096 | 8 | ≤2 compactions |
| 2048 | 8192 | 13 | ≤6 compactions |

*Table 9.* AIME performance (pass@1, averaged over 4 runs) with and without mid-trajectory compaction. *Effective sequence length* is the total number of decoded tokens, while *physical sequence length* is the maximum physical size of the KV cache. Each compaction reduces the size of the entire KV cache (including previously compacted portions) by 50%, so ≤2 and ≤6 compactions correspond to approximately 2× and 4× more decoded tokens, respectively. For implementation simplicity, we do not enable non-uniform compaction or on-policy queries. We use AM-HighestAttnKeys-fast. We believe more tuned methods (e.g. with higher compaction ratios, compacting only the most recently generated tokens, etc.) may improve results.

We conduct a proof-of-concept evaluation of online compaction on **AIME 2025** with Qwen3-4B. For each problem, we cap the *physical* sequence length during reasoning to a fixed budget (Phys Len). Whenever the model reaches this budget, we compact the entire context *except* for the most recent 20 tokens, and then continue decoding until reaching a target *effective* decoding length (Eff Len). We compare against standard decoding to the same effective length without any compaction. To control token budgets, when the model reaches the maximum context length we inject `\nI need to respond with the answer.\n</think>Final Answer:` (following Muennighoff et al. (2025)) and decode the final answer.

Overall, these results suggest that online compaction can effectively decouple reasoning depth from physical context limits. Despite shrinking the KV cache mid-trajectory up to six consecutive times, the model achieves performance comparable to standard decoding at the same effective length, indicating that essential reasoning state is largely preserved. Our study uses a simple uniform compaction scheme and always compacts the entire context, rather than, for example, freezing previously compacted portions or selectively compacting recent tokens. We believe that more careful tuning could enable much greater scaling. We present these results primarily to demonstrate the strong empirical stability of reasoning under repeated mid-trajectory compaction.

Finally, extending online compaction to multi-turn settings, such as compacting long tool call outputs, remains an interesting application for future work.

## F.4. Additional Benchmarks

This section provides results beyond the QASPER and LongBench v2 experiments reported in Section 4.2: the token-merging baseline KVMerger on our default QuALITY setting, and the retrieval-intensive RULER benchmark. Dataset and protocol details for all of these benchmarks are given in Appendix B.2. All tables report accuracy (%) unless stated otherwise; columns give the retained cache size relative to the article tokens, with `0%` the no-context baseline and `100%` the original full cache. The best compaction result in each column is bolded; blank cells denote ratios not evaluated for that method. Across every benchmark, Attention Matching achieves the strongest performance, consistent with our main results.

**KVMerger on QuALITY.** We first add the token-merging baseline KVMerger (Wang et al., 2024) to our default QuALITY setting (Qwen3-4B). KVMerger performs at a similar level to our token-eviction baselines and well below Attention Matching across all ratios.

| Method | 0% | 1% | 2% | 5% | 10% | 20% | 100% |
|---|---|---|---|---|---|---|---|
| Baseline (no ctx / orig.) | 36.8 | | | | | | 71.1 |
| KVMerger | | 43.4 | 49.4 | 55.1 | 60.9 | 65.5 | |
| AM-HighestAttnKeys | | **60.4** | **65.0** | **70.1** | **71.3** | **71.0** | |

*Table 10.* **QuALITY accuracy (%) with KVMerger**, Qwen3-4B, 50 contexts. KVMerger tracks our token-eviction baselines and is well below Attention Matching.

**RULER (retrieval-intensive).** We evaluate over the 6,500 RULER tasks in `kvpress`. Here we use only `AM-HighestAttnKeys-fast` (repeat-prefill queries, no self-study or OMP), our fastest variant, which surpassed KVzip (the prior state of the art on this benchmark) and all leaderboard methods. Numbers for AdaKV, SnapKV, Finch, KeyDiff, TOVA, and PyramidKV are taken from the public `kvpress` leaderboard.

| Method | 0% | 5% | 12.5% | 100% |
|---|---|---|---|---|
| *Llama-3.1-8B-Instruct* | | | | |
| Baseline (no ctx / orig.) | 2.6 | | | **95.7** |
| PyramidKV | | | 27.2 | |
| TOVA | | | 46.4 | |
| KeyDiff | | | 55.7 | |
| Finch (query-aware) | | | 66.7 | |
| SnapKV (query-aware) | | | 67.7 | |
| AdaKV | | | 68.7 | |
| KVzip | | 46.6 | 92.3 | |
| AM-HighestAttnKeys-fast | | **83.3** | **95.1** | |
| *Qwen3-4B* | | | | |
| Baseline (no ctx / orig.) | 3.1 | | | **93.9** |
| KVzip | | 43.9 | 89.1 | |
| AM-HighestAttnKeys-fast | | **52.8** | **90.2** | |

*Table 11.* **RULER accuracy (%)** over 6,500 `kvpress` tasks. Even our fastest variant outperforms KVzip and all leaderboard methods at every evaluated ratio.

# G. Additional Analysis

## G.1. Instruction Following and Failure Modes

Because Attention Matching supports *selective* compaction, critical tokens (e.g., system prompts or chat-template tokens) can be left uncompacted; our experiments already keep these tokens fixed. Since we use identical prompts for original-cache and compacted-cache generation, any degradation in alignment or instruction following would surface in our QA and reconstruction metrics. Qualitatively, compacted and original generations differ little except at very high compaction ratios, where compacted outputs tend to lack specific information (e.g., responding "the text does not mention. . .") while remaining coherent—a graceful, low-risk failure mode.

## G.2. Compatibility with Attention Mechanisms

Attention Matching operates per layer and per KV-head, so it composes naturally with hybrid architectures: our Gemma-3 experiments compact only the global-attention layers of a sliding-window model, and the per-head formulation is directly compatible with GQA under tensor parallelism (each rank owning a KV head). It is, however, incompatible with multi-head latent attention (MLA), which jointly caches a low-rank latent rather than separate keys and values; we note that compaction along the sequence dimension is complementary to MLA's goal of shrinking the per-token cache and can yield substantially larger memory reductions on GQA models.

## G.3. Disaggregated and Asynchronous Compaction

Attention Matching computes the compacted cache as a function of an existing KV cache, with no gradient updates or modification to the serving model. Compaction can therefore be performed off the request critical path, analogous to prefill/decode disaggregation. A separate compaction worker reads a snapshot of the prefix's keys and values and computes the compacted cache; the serving engine continues to answer from the original prefix until compaction completes, then swaps the prefix at a token boundary (e.g., between turns) and reclaims the original memory. Under this scheme compaction adds no user-facing latency—its cost is background compute and a KV-cache transfer.

