# OpenReview forum: "Fast KV Compaction via Attention Matching"
_ICML.cc/2026/Conference — ICML 2026 regular_

### Official Review · Reviewer_NJjR · 2026-03-05

**Soundness:** 3
**Presentation:** 2
**Significance:** 3
**Originality:** 3
**Overall Recommendation:** 4
**Confidence:** 3

**Summary:**

This paper propose Attention Matching (AM), a novel approach for KV compaction in the latent space. Unlike traditional token-eviction methods  that simply drop tokens, or gradient-based latent methods that are computationally expensive, AM treats compaction as a signal reconstruction problem. It optimizes a small set of compressed KV pairs to match the original attention outputs and mass using a fast, closed-form solver.

**Compliance With Llm Reviewing Policy:**

Affirmed.

**Final Justification:**

The author's rebuttal has adequately addressed my concerns.

**Key Questions For Authors:**

Could the authors include a comparison on the RULER benchmark?

**Limitations:**

yes

**Strengths And Weaknesses:**

Strengths：
- The proposed method outperforms existing baselines under extreme compression constraints, maintaining high fidelity to the original model's behavior.
- The paper provides a comprehensive set of experiments across multiple datasets and models, offering robust empirical evidence to support its claims.

----

Weaknesses：

- The paper lacks evaluation on widely recognized long-context benchmarks such as RULER. Consequently, it is difficult to assess the performance of the proposed method on retrieval-intensive tasks (e.g., "Needle-in-a-Haystack").

---

> ### Author Rebuttal · Authors · 2026-03-31
>
> We provide a comparison to RULER below, as well as additional results on LongBench v2 and QASPER in our responses to reviewers nEX5 and acG4 respectively. These results further support our main claim that Attention Matching is highly effective across a broad range of datasets and task types.
>
> We appreciate the reviewer’s suggestion and believe the added RULER comparison meaningfully strengthens the paper.
>
> ## Additional result: RULER
>
> Evaluation was done over 6500 RULER tasks from `kvpress` (*Expected Attention: KV Cache Compression by Estimating Attention from Future Queries Distribution*; Devoto 2025).
>
> KVzip is the prior state of the art on this benchmark. Results for additional methods (AdaKV, SnapKV, Finch, KeyDiff, TOVA, and PyramidKV) on `Llama-3.1-8B-Instruct` were taken from the `kvpress` leaderboard.
>
> **Llama-3.1-8B-Instruct**:
>
> | Method                  | 0% (no ctx) | 5%        | 12.5%     | 100% (original) |
> | ----------------------- | ----------- | --------- | --------- | --------------- |
> | Baseline                | 2.6%        |           |           | **95.7%**       |
> | PyramidKV               |             |           | 27.2%     |                 |
> | TOVA                    |             |           | 46.4%     |                 |
> | KeyDiff                 |             |           | 55.7%     |                 |
> | Finch (query-aware)     |             |           | 66.7%     |                 |
> | SnapKV (query-aware)    |             |           | 67.7%     |                 |
> | AdaKV                   |             |           | 68.7%     |                 |
> | KVzip                   |             | 46.6%     | 92.3%     |                 |
> | AM-HighestAttnKeys-Fast |             | **83.3%** | **95.1%** |                 |
>
> **Qwen3-4B**:
>
> | Method             | 0% (no ctx) | 5%        | 12.5%     | 100% (original) |
> | ------------------ | ----------- | --------- | --------- | --------------- |
> | Baseline           | 3.1%        |           |           | **93.9%**       |
> | KVzip              |             | 43.9%     | 89.1%     |                 |
> | AM-HighestAttnKeys-Fast |             | **52.8%** | **90.2%** |                 |
>
> Our fastest Attention Matching variant (`AM-HighestAttnKeys-Fast`) is able to outperform KVzip and all prior methods on this benchmark.
>
> We emphasize that we believe our other evals (QuALITY, LongHealth, QASPER, and LongBench v2) represent real-world data better than this generalized needle-in-a-haystack test, and AM is able to achieve even greater improvements over these baselines on those benchmarks.

---

> > ### Author Rebuttal · Reviewer_NJjR · 2026-04-01
> >
> > I appreciate the authors' efforts in addressing my comments. All my concerns have been resolved.

---

> > > ### Author Response · Authors · 2026-04-02
> > >
> > > Hi there, thanks for engaging in the discussion process!
> > >
> > > If all of your concerns have been addressed, would you consider updating your score? If not, could you let us know if there are additional questions/weaknesses that we can address during the discussion period?
> > >
> > > Thanks!

---

### Official Review · Reviewer_rWCm · 2026-03-11

**Soundness:** 3
**Presentation:** 3
**Significance:** 3
**Originality:** 3
**Overall Recommendation:** 4
**Confidence:** 4

**Summary:**

The paper addresses KVCache compression from a novel perspective: the factors that most affect the generation quality of large models are actually the Attention Output and Attention Mass, both of which are related to the KVCache. Therefore, by fitting these two quantities, KVCache compression can be guided while preserving output quality. This approach has the potential to overcome the degradation in generation quality caused by previous methods that directly compress the KVCache. Experimental results show that the proposed method surpasses the Pareto frontier under the selected baselines.

**Compliance With Llm Reviewing Policy:**

Affirmed.

**Final Justification:**

Since my concerns have been fully addressed, I maintain my positive stance on this paper.

**Key Questions For Authors:**

1. How well does this method generalize across tasks? Could the authors provide results on datasets such as LongBench or PassKey?

2. Additional experimental details: sensitivity of the reference-query approach (i.e., how sensitive is the method to the number and distribution of reference queries used in the attention-matching process?) and end-to-end latency measurements.

3. Performance under different attention mechanisms: Is this method compatible with native low-rank attention (e.g., DeepSeek MLA)? If it is difficult to experimentally verify due to limitations in accessing open-source implementations of attention, a theoretical or analytical discussion would also be helpful.

**Limitations:**

Yes.

**Strengths And Weaknesses:**

Strengths:

1. Technically rigorous: The core idea, using a compact implicit representation to match the attention output produced by the full KVCache has the potential to preserve more information than methods that directly compress the KVCache under equivalent conditions. From an implementation perspective, the authors formulate the compression problem as a sparse approximation of the key vectors combined with a least-squares fitting of the value vectors, which is a reasonable approach. The use of OMP provides a theoretically grounded method for greedy sparse selection.

2. Somewhat innovative: The proposed train-free, Attention Output-guided KVCache compression method offers promising directions for future research. In particular, end-to-end applications in agent scenarios introduce new challenges. For example, how question-answering scenarios might affect the method’s effectiveness.

Weaknesses:

1. Limited evaluation scenarios: The experiments are mainly conducted on QuALITY and LongHealth. I am concerned that steps in the method, such as collecting “reference queries,” may be sensitive to the evaluation scenario. For example, whether the method’s accuracy could degrade in tasks like mathematical reasoning, code generation, or text retrieval ?

---

> ### Author Rebuttal · Authors · 2026-03-31
>
> We thank the reviewer for their positive feedback and comments. We address each weakness and question.
> ## 1. Limited Evaluations
> We provide additional evaluation on LongBench v2, RULER, and QASPER in our responses to reviewers nEX5, NJjR, and acG4, respectively. The results consistently support our main claim that Attention Matching (AM) performs significantly better than the current state of the art across diverse datasets and models. We also note our evaluation on AIME, a reasoning benchmark, in Appendix F.3.
>
> These results use the same approach and prompts for producing reference-queries. We believe future work that eliminates dependencies on producing queries is promising as well.
> ## 2. Experimental Details on reference queries and latency
> We provide an analysis of reference query approaches in Appendix D. We find the method largely insensitive to choice of reference-query constriction, provided that they are produced via the model’s own forward pass, and in sufficient quantity. Self-study generally does better than repeat-prefill and context-prefill, though we use repeat-prefill for its generality and speed in several experiments (e.g. AIME, RULER, QuALITY) under the name `AM-HighestAttnKeys-Fast`.
>
> We provide latency numbers in Table 1. The main cost is from OMP key selection, if used, and self-study query generation, if used. We note that our query-generation approach can be significantly sped up by optimizing our prefill and decode setups. Finally, in real-world deployments, compaction can be done asynchronously in a disaggregated setup, eliminating any added latency for users.
> ## 3. Compatibility with different attention mechanisms
> The method is incompatible with MLA as it depends on separately-cached keys and values. We note that MLA is primarily used to reduce KV cache memory usage, but Attention Matching on top of GQA is able to achieve much greater memory reduction through compaction along the sequence dimension.
>
> Attention Matching is compatible with attention layers in hybrid models as we compact at a per-layer, per-head level. Indeed, our experiments on Gemma (which uses SWA) show that Attention Matching performs well in this setting. Additionally, since the method works at a per-KV-head level, it is well compatible with GQA under tensor-parallelism (with each rank owning a KV head), allowing for gains with multi-GPU inference.
>
> We will provide discussion on this in the revision.
>
> We thank the reviewer again for these thoughtful questions.

---

> > ### Author Rebuttal · Reviewer_rWCm · 2026-04-02
> >
> > Thank you for your detailed response. My concerns have been fully addressed.

---

> > > ### Author Response · Authors · 2026-04-02
> > >
> > > Thanks for reading our rebuttal and engaging in the discussion.
> > >
> > > Since all of your concerns have been addressed, would you consider updating your score? Otherwise, please let us know if you have any other questions or would like to see additional experiments.
> > >
> > > Thanks.

---

### Official Review · Reviewer_nEX5 · 2026-03-13

**Soundness:** 2
**Presentation:** 2
**Significance:** 2
**Originality:** 3
**Overall Recommendation:** 3
**Confidence:** 4

**Summary:**

This paper introduces Attention Matching, a highly efficient method for compressing the KV cache in large language models to solve the memory bottlenecks associated with long-context processing. Unlike prior latent-space methods that require hours of expensive gradient-based optimization , Attention Matching explicitly engineers a smaller, compacted set of keys, values, and per-token scalar biases to reproduce the attention outputs and overall attention mass of the original, full-sized cache. By breaking this objective down into subproblems that can be solved quickly using standard linear algebra techniques (like Ordinary and Nonnegative Least Squares) , the proposed approach achieves up to 50x cache compaction in a matter of seconds. Evaluated on long-context benchmarks like QuALITY and LongHealth, Attention Matching establishes a new speed-quality Pareto frontier, significantly outperforming token eviction and summarization baselines while matching the accuracy of much slower, state-of-the-art latent optimization techniques.

**Compliance With Llm Reviewing Policy:**

Affirmed.

**Final Justification:**

The experimental results are impressive, and I have increased my score accordingly from 2 to 3. However, I remain personally uncertain about whether this approach can operate reliably in a production environment, what potential risks may exist, and what qualitative changes occur. These concerns are the basis for my final score.

**Key Questions For Authors:**

please refer to the weakness above.

**Limitations:**

yes (in future work section)

**Strengths And Weaknesses:**

**Strengths**

* The ablation study sufficiently demonstrates the effectiveness of the proposed method.
* The proposed approach achieves superior performance compared to existing methods at high compression ratios.
* In particular, it successfully reduces the excessive optimization cost of the existing cartridge-based approach.

**Weaknesses**

* Although the paper is empirically motivated, the evaluation is not comprehensive. The evaluation domains and tasks are limited to QA tasks on QuALITY and LongHealth. It would be helpful to include evaluations on more comprehensive domains such as those in LongBench v2 (e.g., code, tables, multilingual).
* In addition, beyond simple QA, evaluating generation quality on tasks more aligned with real LLM usage (e.g., reasoning, agentic coding, summarization) would provide a better understanding of the practical significance of the proposed method.
* The proposed approach optimizes new KV representations using a method that differs from standard LLM forwarding. Since this approach is not aligned with typical LLM training procedures, careful consideration is required for real-world use, and there is no clear guarantee that alignment or instruction following will remain intact. More in-depth analysis, including failure analysis and discussion of how the method can be safely applied in real applications, would strengthen the paper.
* In Figure 3, some colors across graphs are difficult to distinguish at a glance. Using different marker shapes and more clearly distinguishable colors could improve readability.
* How does the proposed method compare with approaches that compress the KV cache efficiently at inference time using adapters, such as *“Compressed Context Memory For Online Language Model Interaction”* (ICLR 2024)?

---

> ### Author Rebuttal · Authors · 2026-03-31
>
> We thank the reviewer for their feedback. We address each weakness below and will incorporate discussion and several additional experiments in the revision.
> ## 1. Evaluation Scope and Additional Benchmarks
>
> We agree that broader evaluation would strengthen the paper. We include additional experiments across 3 more datasets:
> - LongBench v2: covering code, tables, multilingual tasks, and more.
> - QASPER: generative QA with different structure and reasoning requirements.
> - RULER: generalized needle-in-a-haystack evals, multi-hop tracing, etc.
>
> Our response to Reviewer NJjR contains RULER results, and to Reviewer acG4 contains QASPER results and additional comparisons with KVmerger. LongBench v2 results are given below.
>
> These results consistently support our main claim: Attention Matching (AM) achieves results *significantly* better than the current state of the art across diverse datasets and models.
>
> ## 2. Lack of evaluation on generation-style tasks
> We agree that evaluating on generation tasks in addition to MCQ tasks is important, so we include QASPER.
>
> We also note that we always evaluate with CoT and we measure the perplexity against generations that were produced using the full original cache in Appendix F.2. We show that our method not only has high QA accuracy, but high token-level fidelity against generations that were produced without compaction.
>
> We also note that appendix F.3 includes a reasoning-focused evaluation (AIME) with online compaction. However, agentic coding evaluations are currently infeasible for academic-scale models and setups. We will clarify this limitation explicitly.
>
> ## 3. Concerns about instruction following and need for deeper failure analysis
> We appreciate this important point and will expand the discussion. Our method supports selective compaction, allowing critical tokens (e.g., system prompts) to remain un-compacted. In fact, our current evaluation keeps chat-template tokens fixed during compaction to ensure fair comparison and preserve behavior.
>
> We do not modify our prompts or other aspects of our evals between original-cache generation and compacted generation, and any major differences in behavior would yield a difference in eval outcomes. We hypothesize that improving fidelity with original-cache generations encompasses preserving alignment and instruction following.
> Qualitatively, there is little difference between generations from the compacted cache and generations from the original cache, except at very high compaction ratios, where compacted generations exhibit a lack of information, such as responding with “The text does not mention…,” though remaining coherent. We can provide qualitative examples in the appendix.
>
> ## 4. Figure 3 readability
> We thank the reviewer for this suggestion. We will use more distinguishable colors so that these methods can be distinguished better while keeping the figure readable.
>
> ## 5. Comparison to inference-time adapter methods
> We will expand the related work discussion to discuss tradeoffs of methods such as *Compressed Context Memory*.
>
> Key differences:
> - Adapter-based approaches typically require gradient-based optimization at inference time and introduce additional parameters, though they have been shown to have a higher capacity, e.g., in *Understanding LoRA as Knowledge Memory: An Empirical Analysis* (Back 2026).
> - Attention Matching requires no gradient descent and operates directly in KV-cache space, making it faster and significantly easier to serve at scale.
>
> ## Additional Result: LongBench v2
>
> Model: `Qwen3-4B-Instruct-2507`
>
> Since LongBench v2 was difficult for the models we use, we first ran the original-cache baseline and the no-context baseline on the full dataset, and then selected the subset of questions that were answered correctly by the model with the full context and incorrectly without the context. This produced a set of 47 contexts (max 97k tokens, average 42k tokens) that allowed evaluation with lower variance. We evaluate all methods (including re-evaluating `no ctx` and `original`) on this subset.
>
> | Method             | 0% (no ctx) | 1%        | 2%        | 5%        | 10%       | 20%       | 100% (original) |
> | ------------------ | ----------- | --------- | --------- | --------- | --------- | --------- | --------------- |
> | Baseline           | 0.170       |           |           |           |           |           | **0.723**       |
> | Summarization      |             |           | 0.234     | 0.298     | 0.447     |           |                 |
> | H2O+               |             | 0.106     | 0.213     | 0.362     | 0.447     | 0.617     |                 |
> | KVMerger           |             | 0.192     | 0.213     | 0.340     | 0.447     | 0.489     |                 |
> | KVzip              |             | 0.170     | 0.213     | 0.362     | 0.489     | 0.660     |                 |
> | AM-HighestAttnKeys |             | **0.617** | **0.596** | **0.660** | **0.702** | **0.681** |                 |

---

> > ### Author Rebuttal · Reviewer_nEX5 · 2026-04-03
> >
> > Thank you for your response. The experimental results are impressive, and I have increased my score accordingly from 2 to 3. However, I remain personally uncertain about whether this approach can operate reliably in a production environment, what potential risks may exist, and what qualitative changes occur. These concerns are the basis for my final score.

---

> > > ### Author Response · Authors · 2026-04-04
> > >
> > > Thank you for recognizing the strong experimental results and raising your score to 3.
> > >
> > > We understand your remaining concern regarding reliability and deployment in production environments, and we would like to clarify why we are optimistic about this aspect.
> > >
> > > We believe the method integrates naturally into modern LLM serving systems. In production inference, it is common to disaggregate prefill and decode across different systems, with the prefill server sending a KV cache to the decode server. We believe this architecture can be augmented with a latent compaction step: the prefill server can extract not only keys and values, but also the queries needed for Attention Matching, and then send *compacted* keys and values to the decode server instead of the full cache, when enabled.
> > >
> > > As we noted in our response to reviewer rWCm, this compaction can be performed asynchronously in such a disaggregated setup, with the compacted portion hot-swapped in once the prefill/compaction server finishes and sends over the compacted cache. In this setting, the end user would see no added latency. We also note that this is not something that can be done as naturally with traditional summarization-based compaction, since summarization is not designed to approximate the behavior of the original cache.
> > >
> > > More broadly, we expect this latent compaction setting to be useful in settings where KV-cache memory is the bottleneck and absolute maximum performance is not required. We believe this is a common regime in practical LLM serving. We are currently working on integrating latent context compaction via Attention Matching into a popular open-source inference engine with support of the maintainers.
> > >
> > > We agree that expanding the discussion of production deployment, failure modes, and practical risks would strengthen the paper, and we are happy to do so in revision.
> > >
> > > If there are particular concerns about deployment that would be most helpful to address, we would be glad to clarify them.

---

### Official Review · Reviewer_acG4 · 2026-03-13

**Soundness:** 2
**Presentation:** 3
**Significance:** 2
**Originality:** 3
**Overall Recommendation:** 4
**Confidence:** 3

**Summary:**

This paper proposes Attention Matching, a fast KV cache compaction method operating in latent space. The idea is to select a small subset of representative key vectors, modify the value vectors to minimize the distribution shift caused by compacting the tokens, and compute an attention bias to compensate for the lost attention mass. The compaction process is decomposed into subproblems with explicit closed-form solutions. Empirical evaluations show that Attention Matching achieves the Pareto frontier under 50× KV cache compaction.

**Compliance With Llm Reviewing Policy:**

Affirmed.

**Final Justification:**

The rebuttal addressed my concerns through extensive additional experiments. Therefore, I have raised the scores. I believe the paper adds value to the community both in terms of suggesting a novel approach, and showing empirical performance improvements under small KV cache budgets.

**Key Questions For Authors:**

[Q1] How does Attention Matching perform on free-form generation tasks such as Qasper, or on needle-in-a-haystack benchmarks such as RULER?

[Q2] How does Attention Matching compare with KV merging approaches (such as KVMerger), which appear to share a similar underlying paradigm?

**Limitations:**

yes

**Strengths And Weaknesses:**

**Strengths**

[S1] The paper is clearly written and easy to follow.

[S2] Attention Matching achieves strong empirical performance on multi-hop QA tasks while being significantly faster than gradient-optimization-based approaches such as Catridges.

[S3] Optimizing the KV cache using closed-form optimization is a novel and interesting idea.

**Weaknesses**

[W1] The evaluation is focused on multiple-choice question answering tasks. It remains unclear whether the proposed KV cache compaction method can maintain precise information and achieve strong performance on tasks such as generative QA (e.g., Qasper) or needle-in-a-haystack benchmarks such as RULER.

[W2] Methodologically, Attention Matching appears more closely related to KV cache merging approaches. For example, KVMerger has explored methods that fix a small set of key vectors and aggregate value vectors, which aligns with the high-level paradigm of Attention Matching. However, these works are only briefly mentioned in the related work section. It would be helpful to discuss them in more detail, including the core differences from previous literature.

[W3] While faster than Catridges, Attention Matching still consumes a nontrivial amount of time for the best configurations, which may limit its practical applicability.

[1] Dasigi et. al., A Dataset of Information-Seeking Questions and Answers Anchored in Research Papers, NAACL 2021

[2] Hsieh et. al., RULER: What's the Real Context Size of Your Long-Context Language Models?, COLM 2024

[3] Wang et. al., Model Tells You Where to Merge: Adaptive KV Cache Merging for LLMs on Long-Context Tasks, 2024

---

> ### Author Rebuttal · Authors · 2026-03-31
>
> We thank the reviewer for their feedback. We address each weakness below.
>
> ## [W1] / [Q1] The evaluations only focus on MCQs
> We chose our primary datasets, QuALITY and LongHealth, deliberately because they required reasoning over information-dense real-world long contexts, while also being highly controlled (pairing each context with 15-100 associated MCQs).
>
> However, we agree that showing strong results on non-MCQ evals would strengthen the paper. We thank the reviewer for suggesting QASPER and RULER. We evaluate on QASPER below and RULER in our response to reviewer NJjR. We also evaluate on LongBench v2 in our response to reviewer nEX5. Results show that Attention Matching is very strong across all of these domains.
>
> ## [W2] / [Q2] Comparison to KVMerger and other related work
> Thank you for bringing up this similarity.
>
> KVMerger groups consecutive similar keys (measured by cosine similarity) into clusters, and merges each cluster into a representative KV pair by weighted averaging. Our method instead globally subsets keys and then refits only their values and attention biases to “merge” with the global set of unchosen keys. We will add this discussion of KVMerger and more discussion of prior methods in our Related Work.
>
> Based on your suggestion we additionally evaluated KVMerger on QuALITY, QASPER, and LongBench v2, with results below and in response to reviewer nEX5.
>
> ## [W3] Attention Matching still takes nontrivial time
> The main cost is from OMP key selection, if used, and self-study query generation, if used. Note that we present a Pareto frontier of methods in Figure 1, including those that are both fast and high-performing.
>
> We also note that our query-generation approach can be *significantly* sped up by optimizing our prefill and decode setups, and that no quantization was attempted in this work as it was not the focus of the paper. We are confident that several optimizations can make the method as fast as summarization in deployment. Finally, in real-world deployments, compaction can be done asynchronously in a disaggregated setup, eliminating any added latency for users.
>
> ## Additional Result: KVMerger
> Dataset: QuALITY
> Model: `Qwen3-4B`
>
> | Method             | 0% (no ctx) | 1%        | 2%        | 5%        | 10%       | 20%       | 100% (original) |
> | ------------------ | ----------- | --------- | --------- | --------- | --------- | --------- | --------------- |
> | Baseline           | 36.8%       |           |           |           |           |           | **71.1%**       |
> | KVMerger           |             | 43.4%     | 49.4%     | 55.1%     | 60.9%     | 65.5%     |                 |
> | AM-HighestAttnKeys |             | **60.4%** | **65.0%** | **70.1%** | **71.3%** | **71.0%** |                 |
>
> This eval corresponds to the top left box of Figure 3. We provide the baseline results and our results, along with the new KVMerger results. KVMerger performs at a similar level to our token eviction baselines, substantially lower than Attention Matching.
>
> ## Additional Result: QASPER
>
> QASPER; Average F1 score, n = 562 contexts, 2010 questions
>
> **Qwen3-4B**:
>
> | Method             | 0% (no ctx) | 1%        | 2%        | 5%        | 10%       | 20%       | 100% (original) |
> | ------------------ | ----------- | --------- | --------- | --------- | --------- | --------- | --------------- |
> | Baseline           | 0.104       |           |           |           |           |           | **0.444**       |
> | Summarization      |             |           |           | 0.219     | 0.283     |           |                 |
> | H2O+               |             | 0.188     | 0.243     | 0.310     | 0.350     | 0.390     |                 |
> | KVMerger           |             | 0.171     | 0.215     | 0.260     | 0.303     | 0.356     |                 |
> | KVzip              |             | 0.127     | 0.124     | 0.220     | 0.411     | **0.438** |                 |
> | AM-HighestAttnKeys |             | **0.338** | **0.373** | **0.416** | **0.428** | 0.435     |                 |
>
>
> **Llama-3.1-8B-Instruct**:
>
> | Method             | 0% (no ctx) | 1%        | 2%        | 5%        | 10%       | 20%       | 100% (original) |
> | ------------------ | ----------- | --------- | --------- | --------- | --------- | --------- | --------------- |
> | Baseline           | 0.105       |           |           |           |           |           | **0.456**       |
> | Summarization      |             |           |           | 0.253     | 0.280     |           |                 |
> | H2O+               |             | 0.199     | 0.243     | 0.301     | 0.348     | 0.372     |                 |
> | KVzip              |             | 0.102     | 0.104     | 0.168     | 0.347     | 0.422     |                 |
> | AM-HighestAttnKeys |             | **0.317** | **0.387** | **0.436** | **0.452** | **0.450** |                 |

---

> > ### Author Rebuttal · Reviewer_acG4 · 2026-04-04
> >
> > Thank you for your efforts in the additional experiments. My concerns are resolved, and I have raised the score accordingly. Good luck!

---

> > > ### Author Response · Authors · 2026-04-06
> > >
> > > Thank you again for the thoughtful review! Please let us know if you have any further questions or ways we could strengthen the paper.

---

### Decision · Program_Chairs · 2026-04-30

**Decision:**

Accept (regular)

**Comment:**

This paper studies a practical problem for LLMs: how to shrink the KV cache for long-context LMs without hurting performance too much. The main idea is to do compaction in latent space by matching attention behavior, instead of relying on token dropping, summarization, or slow gradient-based optimization. Overall, the reviewers found the core idea interesting and technically solid, and they generally agreed that the method gives a strong speed-quality tradeoff. In particular, the results suggest that it works much better than standard token-pruning and summarization baselines at high compaction ratios, while being much faster than Cartridges. Empirically, the paper evaluates on QuALITY and LongHealth, across Qwen3-4B, Llama-3.1-8B, and Gemma-3-12B. The main claim is that Attention Matching improves the speed-quality Pareto frontier: it consistently beats token-pruning and summarization baselines.

The main concern raised in the reviews was evaluation. In the original submission, the experiments were concentrated on QuALITY and LongHealth, and reviewers asked for broader evidence on generation-style and retrieval-heavy settings, as well as clearer discussion of related methods. The author response addressed these concerns with additional results on QASPER, LongBench v2, and RULER, plus clarification of the relation to KVMerger and related approaches. Based on the discussion, most reviewers indicated that their main concerns were resolved.

One reviewer still expressed some concern about production reliability and possible qualitative changes in behavior after compaction. I do not think this remaining concern outweighs the strengths. Overall, I find the contribution technically solid, novel enough for ICML.